# *Drosophila* Fezf coordinates laminar-specific connectivity through cell-intrinsic and cell-extrinsic mechanisms

Jing Peng[1†], Ivan J Santiago[1†], Curie Ahn[1], Burak Gur[2], C Kimberly Tsui[3], Zhixiao Su[1], Chundi Xu[1], Aziz Karakhanyan[1], Marion Silies[2], Matthew Y Pecot[1*]

[1]Department of Neurobiology, Harvard Medical School, Boston, United States; [2]European Neuroscience Institute, Göttingen, Germany; [3]Department of Genetics, Stanford University, Stanford, United States

**Abstract** Laminar arrangement of neural connections is a fundamental feature of neural circuit organization. Identifying mechanisms that coordinate neural connections within correct layers is thus vital for understanding how neural circuits are assembled. In the medulla of the *Drosophila* visual system neurons form connections within ten parallel layers. The M3 layer receives input from two neuron types that sequentially innervate M3 during development. Here we show that M3-specific innervation by both neurons is coordinated by *Drosophila* Fezf (dFezf), a conserved transcription factor that is selectively expressed by the earlier targeting input neuron. In this cell, dFezf instructs layer specificity and activates the expression of a secreted molecule (Netrin) that regulates the layer specificity of the other input neuron. We propose that employment of transcriptional modules that cell-intrinsically target neurons to specific layers, and cell-extrinsically recruit other neurons is a general mechanism for building layered networks of neural connections.
DOI: https://doi.org/10.7554/eLife.33962.001

**\*For correspondence:**
matthew_pecot@hms.harvard.edu

[†]These authors contributed equally to this work

**Competing interests:** The authors declare that no competing interests exist.

## Introduction

Precise neural connectivity underlies the structural organization and function of the nervous system. In the nervous systems of diverse organisms neural connections are arranged into layered networks, wherein each layer is defined by a unique cellular composition and the axonal arborizations of different input neurons are restricted to specific layers. This organizational strategy is thought to optimize the precision of synaptic connectivity and information processing. Layered connections define many regions of the vertebrate nervous system, including the cerebral cortex (mammals), the spinal cord and retina, and connections in the insect optic lobe are also organized in a layer-specific manner. Elucidating how neurons organize into layered networks is crucial for understanding how the precision of neural connectivity is achieved, and may provide key insights into neural function.

Previous studies in both vertebrates and invertebrates have identified molecules that are necessary for layer specificity in particular neuron types (reviewed in) (*Baier, 2013*; *Huberman et al., 2010*; *Sanes and Yamagata, 1999*; *Sanes and Zipursky, 2010*). However, the molecular mechanisms governing the assembly of specific layers remain poorly characterized. For example, how different neurons innervate the same layer, and whether they do so through shared or different mechanisms is not known. To address this, we study layer formation in the *Drosophila* visual system, wherein the cell types that innervate specific layers are well-described and genetically accessible.

In the *Drosophila* optic lobe (*Figure 1A*), visual input converges on the medulla, wherein it is processed within parallel layers. The medulla receives input directly from R7 and R8 photoreceptors, which are UV and blue/green sensitive, respectively, and indirectly from broadly tuned photoreceptors (R1-R6) through second order lamina monopolar neurons L1-L5 (lamina neurons). Photoreceptor

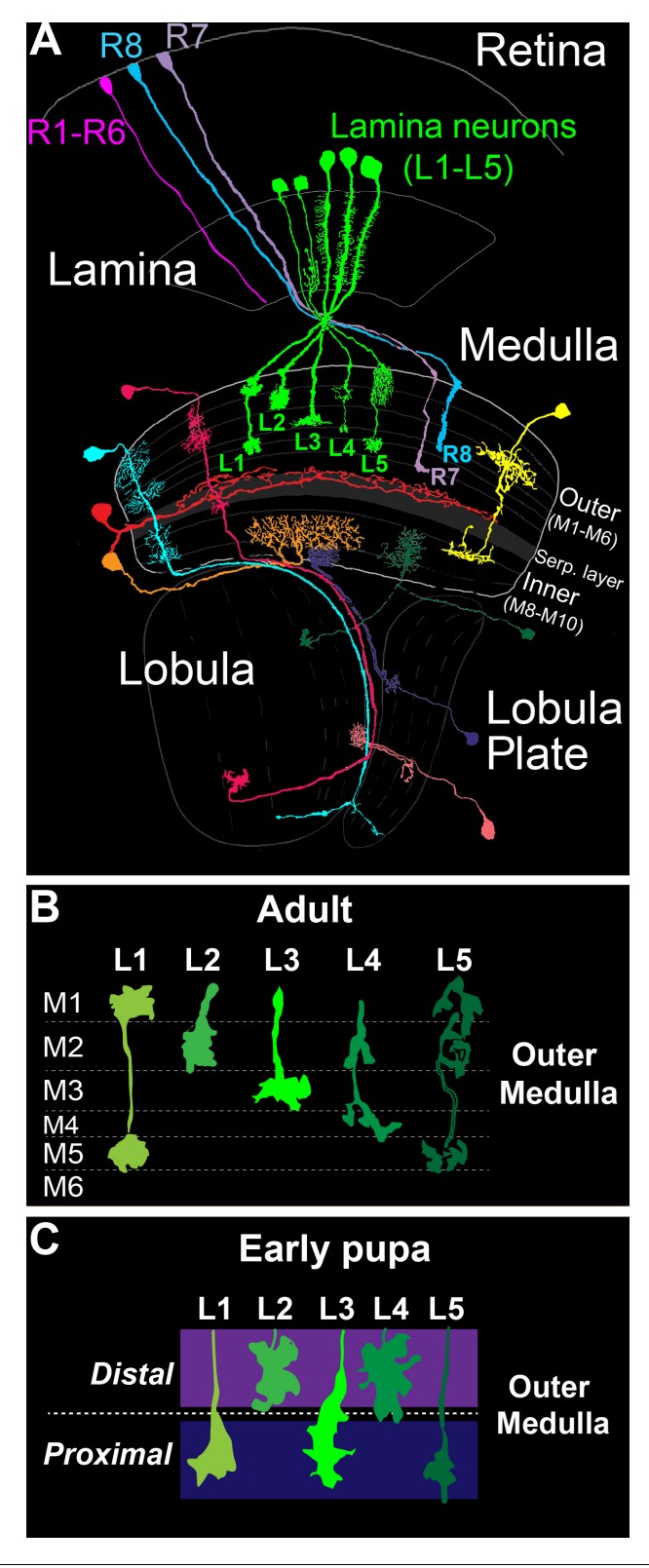

**Figure 1.** The *Drosophila* visual system and lamina monopolar neurons. (**A**) Anatomy of the *Drosophila* visual system (Adapted from *Fischbach and Dittrich, 1989*). The optic lobe comprises four consecutive neuropil regions called the lamina, medulla, lobula and lobula plate. (**B**) Cartoon of lamina neuron axons in adult flies. The nearly mutually exclusive axonal arborizations of lamina neurons help define layers M1-M5. (**C**) Cartoon of lamina neuron

*Figure 1 continued*

growth cones in early pupal development. Prior to innervating discrete layers, lamina growth cones terminate in broad distal or proximal domains within the outer medulla.

DOI: https://doi.org/10.7554/eLife.33962.002

and lamina neuron axons innervate discrete medulla layers and synapse with medulla neurons that process information and transmit signals to higher centers. Information from specific regions of the visual field is processed in modular columnar units, organized perpendicular to the layers. Lamina and photoreceptor axons carrying input from the same point in space converge on targets in the same column. Input from neighboring points in space is processed in neighboring columns, establishing a retinotopic map.

The medulla comprises ten layers (M1-M10) organized into outer (M1-M6) and inner (M8-M10) regions that are divided by tangential processes that form the serpentine layer (i.e. M7) (*Fischbach and Dittrich, 1989*) (*Figure 1A*). The cell bodies of medulla neurons are excluded from the layers, and thus layered connections develop within a dense meshwork of cellular processes. Laminar-specific connections within the inner plexiform layer (IPL) of the vertebrate retina develop in a similar manner. Medulla layers are defined in adult flies by the morphologies of the axon terminals and dendritic branches of particular cell types, which in general overlap completely or not at all (*Figure 1B*). The positions of these processes are largely indicative of the location of synapses, although some neurites do not stratify and form synapses en passant (*Takemura et al., 2013*, *2008*, *2015*). In total, ~40,000 neurons that fall into more than sixty cell types form connections within one or more layers (*Fischbach and Dittrich, 1989*).

Studies of lamina neuron and photoreceptor axon development indicate that medulla layers emerge dynamically from broad domains. The morphologies of L1-L5 axons define layers M1-M5 in the outer medulla of adult animals (*Fischbach and Dittrich, 1989*) (*Figure 1B*). However, in early pupal development the outer medulla is a fraction of its adult size, and lamina growth cones terminate in two broad domains (*Figure 1C*) (*Nern et al., 2008*; *Pecot et al., 2013*). L2 and L4 growth cones terminate in a distal domain of the outer medulla, while L1, L3 and L5 growth cones terminate in a proximal domain. The mechanisms underlying specificity for the distal or proximal domain are not known. Nevertheless, these findings indicate that lamina neurons innervate broad domains prior to segregating into discrete layers. In the mouse IPL, it has been proposed that neurons initially innervate broad domains defined by the complementary expression of repulsive cell surface molecules (*Matsuoka et al., 2011*). Together with studies in the medulla, this suggests that the refinement of broad domains into discrete layers may represent a conserved developmental strategy for constructing layer-specific circuits.

Characterization of the targeting of L3 axons and the axons of R8 photoreceptors to the M3 layer has provided insight into how discrete layers emerge from broad domains. L3 and R8 axons innervate M3 sequentially during development. R8 neurons are born first and project axons that terminate near the distal surface of the medulla (known as M0) (*Ting et al., 2005*). L3 axons within the same column project past R8 axons and terminate within a broad domain in the proximal outer medulla (discussed above) (*Nern et al., 2008*; *Pecot et al., 2013*). Subsequently, L3 growth cones segregate into the developing M3 layer through a stereotyped structural rearrangement that is regulated by mechanisms that are not well understood (*Pecot et al., 2013*). Within M3, L3 growth cones secrete Netrin, which acts locally to regulate the attachment of R8 growth cones within the layer after they extend from M0 (*Akin and Zipursky, 2016*; *Pecot et al., 2014*; *Timofeev et al., 2012*). Thus, the M3 layer develops in part through the stepwise innervation of L3 and R8 axons, and R8 layer specificity depends on a signal from L3 growth cones.

Taken together, these developmental studies point to a model, wherein outer medulla layers emerge in a stepwise manner from broad domains through a precise sequence of interactions between specific cell types. To gain insight into the molecular mechanisms that control stepwise layer assembly we concentrate on assembly of the M3 layer, and in this study focus on L3 neurons. Previously, we found that *Drosophila* Fezf (dFezf)/Earmuff, the *Drosophila* ortholog of the Fezf zinc finger transcription factors in vertebrates (Fezf1/2) (*Hashimoto et al., 2000a*, *2000b*; *Matsuo-Takasaki et al., 2000*; *Pfeiffer et al., 2008*; *Weng et al., 2010*), is selectively expressed in L3

neurons in the lamina (*Tan et al., 2015*). Thus, we reasoned that dFezf might regulate the expression of genes that control cell-specific aspects of L3 development, including broad domain specificity, layer specificity, and synaptic specificity within the target layer.

Interestingly, our genetic analyses of dFezf function in L3 neurons revealed a critical role for dFezf in regulating the M3-specific innervation of both L3 *and* R8 axons. DFezf functions cell autonomously to promote the targeting of L3 growth cones to the proximal domain of the outer medulla, which is essential for L3 layer specificity. We find that members of the dpr cell surface gene family (*Nakamura et al., 2002*) are prominent direct or indirect dFezf targets in L3 neurons, and we propose that dFezf regulates growth cone targeting by controlling the expression of dpr proteins that mediate interactions with target cells in the medulla. In addition, we show that dFezf non-autonomously regulates R8 layer specificity through the activation of *Netrin* expression in L3 neurons. When dFezf function is lost in L3 neurons, L3 and R8 axons innervate inappropriate layers while the dendrites of a common synaptic target (Tm9) innervate the M3 layer normally. As a result, synaptic connectivity with the target cell is disrupted. We conclude that dFezf represents a transcriptional module that constructs M3 layer circuitry by controlling the stepwise layer innervation of specific cell types through cell-intrinsic and cell-extrinsic mechanisms. We propose that the use of such modules to coordinate neural connections within correct layers is a widespread strategy for building laminar-specific circuits.

## Results

### DFezf is selectively expressed in L3 neurons during pupal development and in adult flies

As a first step towards characterizing the function of dFezf in L3 neurons we examined its expression during development. Previously, we showed that dFezf is expressed in L3 neurons at 40 hr after puparium formation (h APF) (*Tan et al., 2015*), when L3 growth cones are in the process of segregating into the developing M3 layer (*Pecot et al., 2013*). To assess if dFezf is also expressed in L3 neurons at other stages of development and determine whether expression remains restricted to L3 neurons, we assessed dFezf immunostaining in the lamina using confocal microscopy. We discovered that dFezf is expressed in L3 neurons from early pupal stages (at least as early as 12 hr APF) through to adulthood, but is not expressed in late third instar larvae when lamina neurons begin to differentiate (*Figure 2A–D*). DFezf expression was not detected in lamina neuron precursor cells or any other cells besides L3 at any stage of development in the lamina. The timing and selectivity of dFezf expression is consistent with dFezf playing an important role in regulating L3 development.

### DFezf is cell autonomously required for L3 layer specificity

To investigate the function of *dFezf* in L3 neurons we used mosaic analysis with a repressible cell marker (MARCM) (*Lee and Luo, 1999*) in conjunction with synaptic tagging with recombination (STaR) (*Chen et al., 2014*). This allowed us to generate single, fluorescently labeled wild type or *dFezf* mutant L3 neurons while simultaneously visualizing the active zone protein Bruchpilot (Brp), expressed from its native promoter (see Materials and methods for detailed description). The *dFezf* null allele used in these experiments (*dFezf¹*) contains a single A to T nucleotide change leading to the substitution of a leucine for a conserved histidine in the third zinc-finger domain (*Weng et al., 2010*). We found that L3 neurons homozygous for *dFezf¹* displayed defects in axon terminal morphology and layer specificity in the medulla. Wild type L3 neurons elaborate stratified axon terminals within the M3 layer (100%) (*Figure 2E,F,J*). The large majority (84%) of the axons of *dFezf¹* L3 neurons terminated inappropriately within layers distal to M3 (i.e. M1-M2) (*Figure 2G,J*) and had abnormally shaped terminals (compare *Figure 2F and G*). The remaining 16% of the axons either terminated in the M3 layer (*Figure 2H,J*) or in more proximal layers (e.g. M4-M6) (*Figure 2I,J*), and these also had terminal-like swellings in M1-M2 (*Figure 2H,I*). Presynaptic Brp puncta were present in mis-targeting L3 terminals (*Figure 2K,L*), suggesting that mutant neurons formed synaptic connections within incorrect layers. However, whether these represent active synapses, and if so whether they are established with appropriate partners, inappropriate partners, or both is unclear. In the lamina, L3 neurons display a unique dendritic projection pattern in which all dendrites project to one side of the neurite (*Figure 2M*). Dendritic processes were largely normal in *dFezf* null neurons

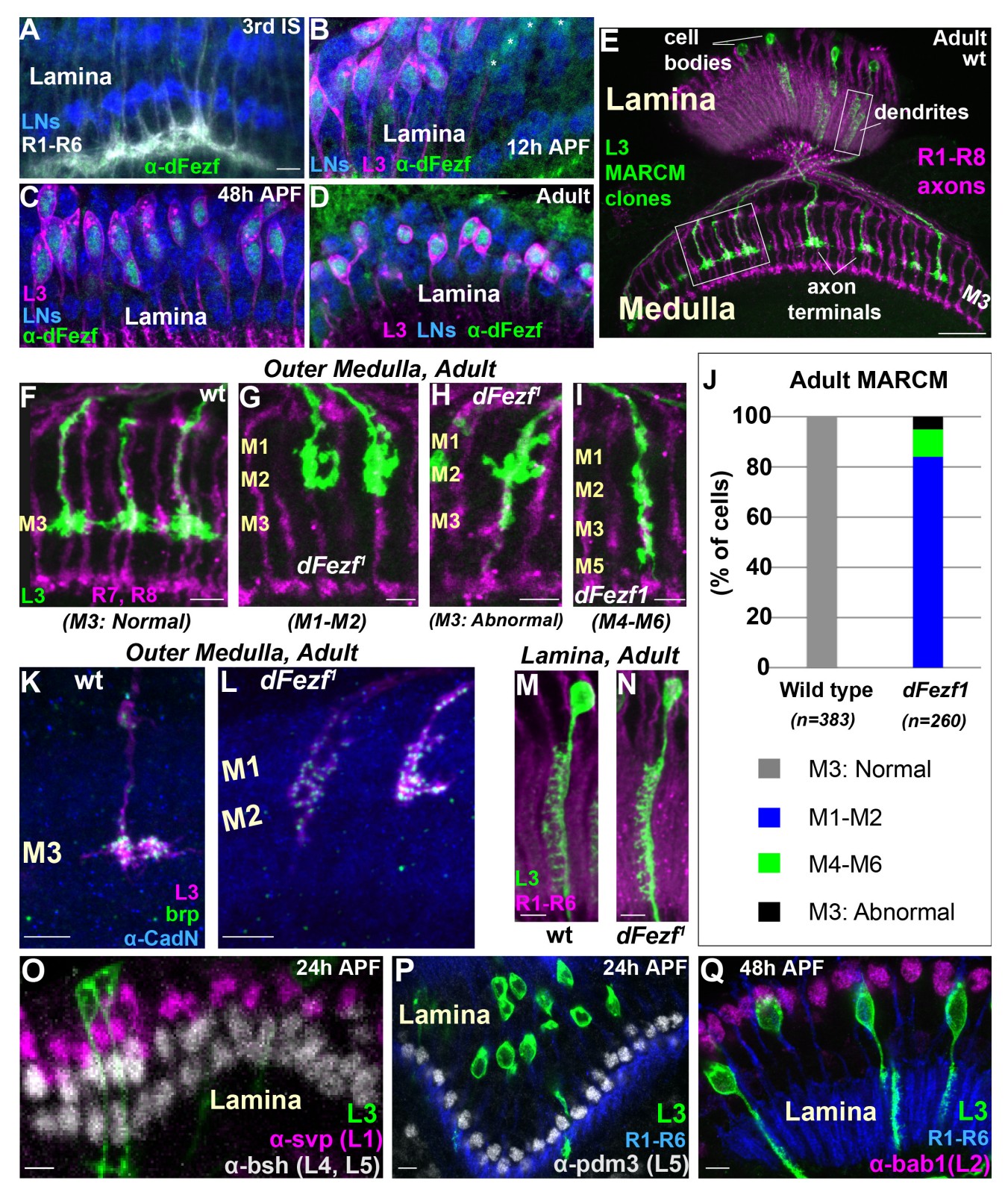

**Figure 2.** *DFezf* is cell autonomously required for L3 layer specificity and axon terminal morphology. (**A–D**) Confocal images showing dFezf protein expression in the lamina from the late 3rd Instar larval stage through to adulthood (newly eclosed flies). DFezf expression (green) was observed through immunohistochemistry using a specific antibody. Elav immunostaining (blue) labels all lamina neurons (LNs). At least five brains were examined per time point. Scale bar = 5 microns and applies to all images. (**A**) DFezf (green) is not expressed in the lamina during the 3rd Instar larval stage. Lamina

*Figure 2 continued on next page*

*Figure 2 continued*

neurons (blue) differentiate in close proximity to R1-R6 axons (white, mAB24B10) forming columnar-like cartridges oriented orthogonally to the lamina plexus (thick white band) comprising R1-R6 terminals. (**B–D**) DFezf (green) is expressed exclusively in L3 neurons in the lamina during pupal development and in newly eclosed adults. L3 neurons (magenta) express myr::GFP (anti-GFP) driven by 9–9 GAL4 (*Nern et al., 2008*). Asterisks in (**B**) indicate dFezf-expressing lamina neurons (green + blue) that are most likely L3 neurons that have yet to turn on GAL4 expression (youngest L3 neurons). (**E–N**) MARCM + STaR experiments in adult flies. See Materials and methods section for a detailed description. Scale bars when shown are five microns with the exception of the scale bar shown in (**E**), which is twenty microns. L3 clones (green) expressed myr::tandem Tomato (tdTomato) and were visualized using a DsRed antibody. (**E**) Broad view (confocal image) of the morphologies of wild type L3 clones (green) in the lamina and medulla, which are indicated by R1-R8 axons (magenta, mAB24B10). Boxed regions indicate examples of the regions shown in confocal images in F-I, M and N. (**F–I**) Confocal images of terminals from wild type or *dFezf¹* L3 neurons (green), within the outer medulla (M1-M6), defined by R7 and R8 axons (magenta, mAB24B10). (**J**) Adult MARCM quantification. N is the total number of neurons counted per genotype. 10 brains were analyzed for each genotype. (**K, L**) Confocal images show L3 clones (magenta) within the outer medulla and, within L3 terminals, Brp (green) expression and localization. Brp expressed from its native promoter within a bacterial artificial chromosome (BAC) was selectively tagged with smFPV5 in L3 MARCM clones, and visualized using an anti-V5 antibody. N-Cadherin (CadN) immunostaining (blue) serves as a neuropil marker. All *dFezf¹* L3 neurons displayed Brp puncta in their terminals. At least five brains were analyzed for each genotype. (**M and N**) Morphologies of dendrites from wild type or *dFezf¹* L3 neurons in the lamina. The lamina is indicated by R1-R6 axons (magenta, mAB24B10). (**O–Q**) MARCM experiments analyzed at 24 or 48 hr APF. *DFezf¹* L3 clones (green) express myr::GFP (9–9 GAL4) and are visualized using a GFP antibody. Co-labeling of L3 somas and transcription factors specific for other lamina neuron subtypes (using specific antibodies) show that the mutant neurons do not express these proteins. (**P and Q**) R1-R6 axons (blue, mAB24B10) demarcate lamina cartridges and the lamina plexus.

DOI: https://doi.org/10.7554/eLife.33962.003

The following source data is available for figure 2:

**Source data 1.** Input data for bar graph *Figure 2J*.

DOI: https://doi.org/10.7554/eLife.33962.004

(compare *Figure 2M and N*). In addition, the mutant L3 neurons did not express transcription factors specific to other lamina neurons (*Figure 2O–Q*) and still expressed L3-specific markers early in development (see *Figure 3*), showing that disrupting *dFezf* did not cause an obvious change in cell fate. However, some L3-specific genes are downregulated in *dFezf¹* L3 neurons, and genes expressed in other lamina neurons become upregulated (see below). Taken together, these findings demonstrate that *dFezf* is cell autonomously required for proper axon terminal morphology and layer specificity, yet dispensable for dendrite formation.

## DFezf is necessary for targeting to the proximal domain of the outer medulla

To determine how disrupting *dFezf* affects L3 axon targeting during development, we compared the axonal morphologies of wild type and *dFezf¹* neurons during pupal development using MARCM. In early pupal development (~24 hr APF), the axons of wild type L3 neurons terminate in the proximal domain of the outer medulla directly above R7 growth cones (*Figure 3A,B,E*). The axons of *dFezf¹* neurons incorrectly terminated within the distal domain of the outer medulla (*Figure 3C–E*). Under these conditions L3 growth cones terminated just underneath the growth cones of R8 photoreceptors (*Figure 3C*), and occasionally terminated slightly more distally so that they completely overlapped with R8 growth cones (*Figure 3D*). By mid-pupal development (~48 hr APF), the growth cones of wild type L3 neurons have segregated into the developing M3 layer (*Figure 3F,G,K*). However, we found that all of the growth cones of *dFezf¹* L3 neurons displayed abnormal morphologies, defects in layer specificity, or both. Most of the abnormal growth cones (67%) exclusively innervated the developing M1 and M2 layers (*Figure 3H,K*). The remaining 33% of the growth cones terminated in the developing M3 layer (*Figure 3I,K*) or in more proximal layers (e.g. M4-M6) (*Figure 3J, K*). However, each of these growth cones also displayed abnormal swellings in distal layers (i.e. the developing M1 and M2 layers) (*Figure 3I,J*, arrowheads). These defects in morphology and layer innervation are consistent with the abnormalities observed in adult flies under these conditions. We observed similar deficits in L3 growth cone targeting in MARCM experiments when we used a different *dFezf* mutant allele (*dFezf²*) that eliminates the entire *dFezf* open reading frame (*Weng et al., 2010*) (*Figure 3—figure supplement 1*). Moreover, defects in growth cone targeting during pupal development, as well as abnormal layer innervation in adult flies caused by *dFezf¹* or *dFezf²* were completely rescued by a single copy of a bacterial artificial chromosome (BAC) containing the *dFezf*

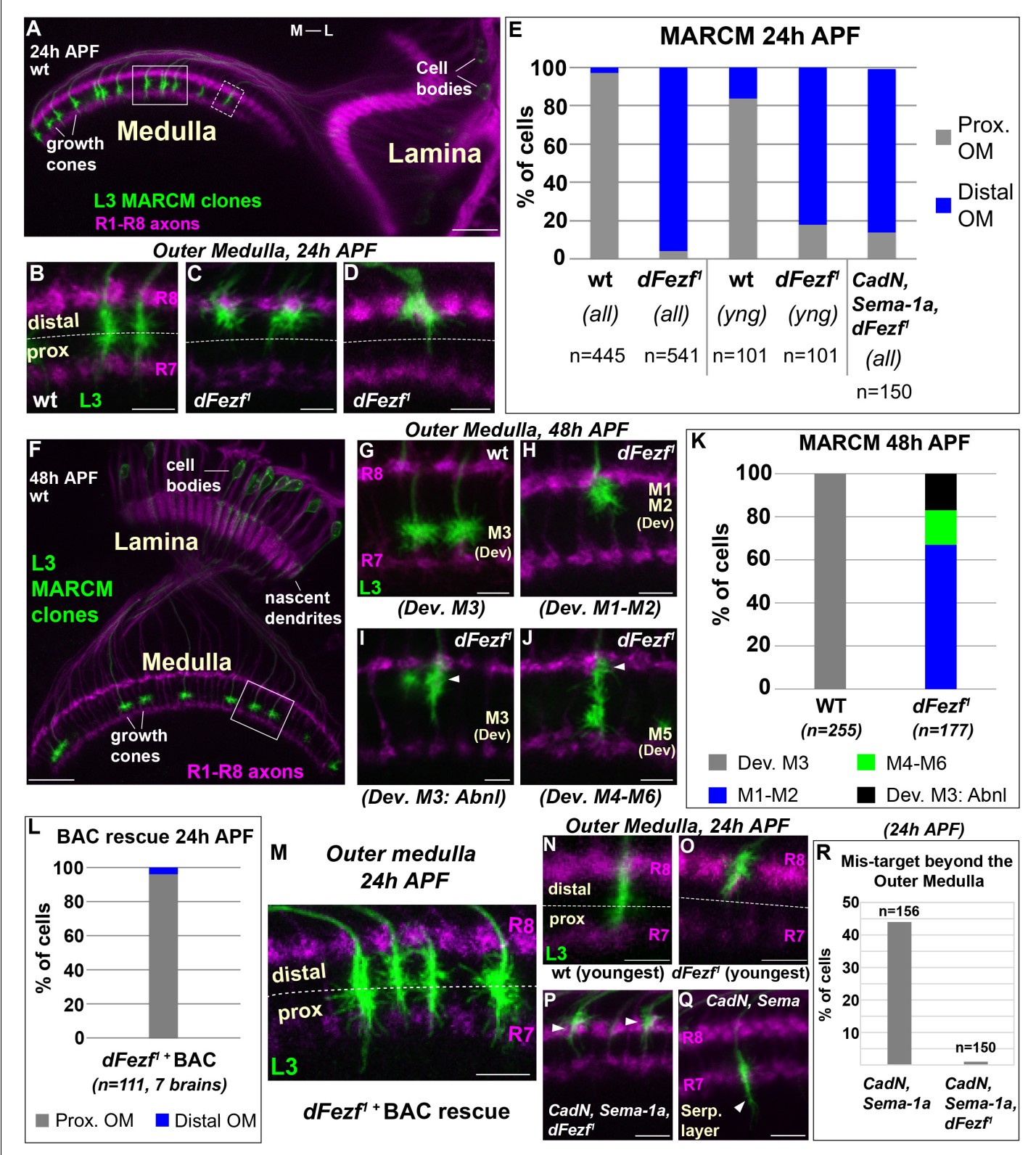

**Figure 3.** *DFezf* is required for the targeting of L3 growth cones to the proximal domain of the outer medulla. (A–R) MARCM experiments analyzed at 24 or 48 hr APF. L3 clones expressed myr::GFP driven by 9–9 GAL4 and were visualized in confocal images using a GFP antibody. In confocal images, mAB24B10 (magenta) was used to label photoreceptor axons, which demarcate the lamina and medulla. Within the medulla, mAB24B10 labels R8 and R7 growth cones, which define the boundaries of the outer medulla. In some images R7 growth cones may appear faint. This reflects normal variations

*Figure 3 continued on next page*

*Figure 3 continued*

in antibody staining between brains of slightly different ages (e.g. ±2 hr). No defects in photoreceptor axon targeting were observed in these experiments. (**A**) Broad view of the morphologies of wild type L3 clones (green) within the lamina and medulla (indicated by R1-R8 axons, magenta [mAB24B10]) at 24 hr APF. The solid boxed region indicates an example of confocal images shown in B-D. The dashed boxed region shows an example of a newly born L3 clone in the most lateral part of the medulla array similar to N and O. M-L indicates medial and lateral. Scale bar = 20 microns. (**B–D, G–J, M–Q**) Confocal images show a zoomed in view of L3 growth cones within the outer medulla. At these stages of development the boundaries of the outer medulla are defined by R7 and R8 growth cones (magenta, mAB24B10). Scale bars = 5 microns. (**E, K, L, R**) Quantification of MARCM experiments. N is the total number of neurons per genotype that were counted. (**E**) WT (seven brains), *dFezf[1]* (12 brains), *CadN, Sema-1a, dFezf[1]* (two brains). (*all*) means that all L3 clones were counted while (*yng*) indicates that the L3 clones present within the most lateral five medulla columns containing R7 growth cones were counted (i.e. youngest L3 clones). Prox. OM = proximal domain of the outer medulla, Distal OM = distal domain of the outer medulla. (**F**) Broad view of the morphologies of wild type L3 clones in the lamina and medulla (indicated by R1-R8 axons, magenta [mAB24B10]) at 48 hr APF. The solid boxed region is an example of regions shown in confocal images in G-J. Scale bar = 20 microns. (**I and J**) Arrowheads indicate abnormal swellings within the developing M1-M2 layers. (**K**) Wild type (six brains), *dFezf[1]* (six brains). (**L**) Prox. OM = proximal domain of the outer medulla, Distal OM = distal domain of the outer medulla. (**P and Q**) Arrowheads indicate the depths within the medulla at which L3 growth cones terminate. The serpentine layer (indicated in Q) lies directly beneath the outer medulla. (**R**) *CadN, Sema-1a* (two brains), *CadN, Sema-1a, dFezf[1]* (two brains).

DOI: https://doi.org/10.7554/eLife.33962.005

The following source data and figure supplements are available for figure 3:

**Source data 1.** Input data for bar graphs *Figure 3E, K, L, R*, and *Figure 3—figure supplement 1*; *Figure 3—figure supplement 2B,2D and 2F*.
DOI: https://doi.org/10.7554/eLife.33962.008

**Figure supplement 1.** *DFezf[2]* L3 neurons show mis-targeting phenotypes similar to those caused by the *dFezf[1]* allele.
DOI: https://doi.org/10.7554/eLife.33962.006

**Figure supplement 2.** A BAC containing the dFezf locus rescues defects in growth cone targeting, layer specificity and axon terminal morphology caused by *dFezf* null mutations.
DOI: https://doi.org/10.7554/eLife.33962.007

locus (*Figure 3L,M*, and *Figure 3—figure supplement 2*). Together, these findings demonstrate that *dFezf* is cell-autonomously required for the targeting of L3 growth cones to the proximal domain of the outer medulla, which may be necessary for the subsequent segregation of the growth cones into the developing M3 layer.

We reasoned that the growth cones of *dFezf* null L3 neurons could directly terminate within the distal domain of the outer medulla, or could initially target correctly to the proximal domain and then retract. Direct mis-targeting would be consistent with a change in axon target specificity, while retraction would suggest a defect in stabilizing growth cone position. To distinguish between these possibilities we took two approaches. First, in MARCM experiments we focused on the L3 clones that had most recently innervated the medulla (i.e. the youngest clones, see dashed box in *Figure 3A*). Unlike the youngest wild type clones, the large majority of the nascent *dFezf[1]* clones (82%) terminated within the distal domain of the outer medulla similar to their older counter parts (*Figure 3E,N,O*), consistent with most of these neurons directly innervating this region. Second, we investigated growth cone targeting when *dFezf* was disrupted in combination with *CadN* and *Sema-1a*. CadN and Sema-1a function synergistically to restrict L3 growth cones to the outer medulla during pupal development, preventing innervation of the serpentine layer and inner medulla (*Pecot et al., 2013*). We hypothesized that if the growth cones of *dFezf[1]* L3 neurons initially target correctly and then retract, then in the absence of CadN and Sema-1a function the growth cones would inappropriately innervate the serpentine layer and inner medulla, as in *CadN* and *Sema-1a* double mutant neurons. Conversely, if the growth cones of *dFezf[1]* L3 neurons directly mis-target to the distal outer medulla, this should prevent mis-targeting beyond the outer medulla caused by the loss of CadN and Sema-1a function, and the triple KO phenotype should resemble the loss of dFezf function only. MARCM experiments revealed the latter to be correct. L3 neurons lacking the function of dFezf, CadN and Sema-1a predominantly innervated the distal domain of the outer medulla in early pupal development (*Figure 3E,P*), similar to L3 neurons lacking dFezf function only (*Figure 3C–E*), although the triple KO neurons more frequently terminated within the R8 growth cone layer. Importantly, the triple KO neurons did not mis-target beyond the outer medulla like *CadN/Sema-1a* double mutant L3 neurons (*Figure 3Q,R*). Combined with our analyses of newly born *dFezf[1]* L3 neurons, these data suggest that L3 neurons lacking dFezf function directly innervate the distal domain of the outer medulla, which is consistent with a switch in broad domain specificity.

## DFezf acts instructively to regulate growth cone targeting to the proximal domain of the outer medulla

DFezf could function permissively or in an instructive manner to regulate the targeting of L3 growth cones to the proximal domain of the outer medulla. To discriminate between these possibilities we performed mis-expression experiments in other lamina neurons. If dFezf function is instructive, then mis-expressing it in L2 or L4 neurons, which normally target to the distal domain of the outer medulla (*Figure 1C*), should cause their growth cones to terminate in the proximal domain. By contrast, as L1 and L5 growth cones normally terminate in the proximal domain (*Figure 1C*), dFezf-mis-expression in these neurons should not affect growth cone targeting. For technical reasons we decided to focus on L2 and L5 neurons. In L5 neurons, we mis-expressed cDNA encoding dFezf-HA using 6–60 GAL4, which has been shown to selectively drive expression in L5 neurons in the lamina (*Nern et al., 2008*). In these experiments all L5 neurons and a small number of medulla and lobula

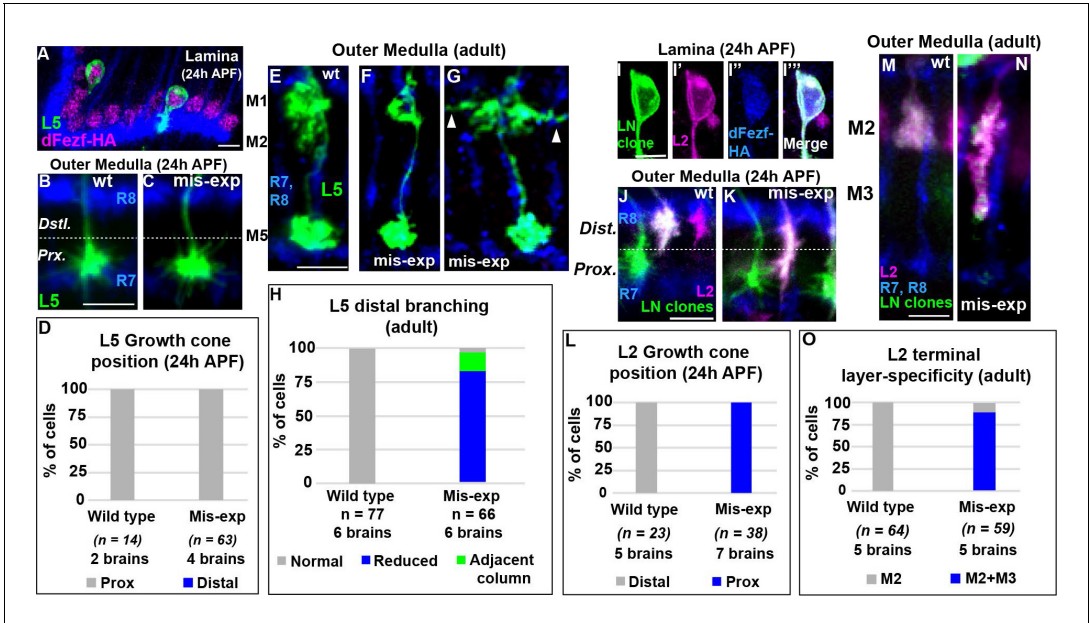

**Figure 4.** DFezf plays an instructive role in regulating growth cone targeting. (A–H) L5 mis-expression experiments. DFezf-HA was expressed in L5 neurons using 6–60 GAL4, and sparse labeling of L5 neurons (myr::GFP, α-GFP) was achieved using the FlpOut method (see Materials and methods for detailed description). (A) DFezf-HA (magenta) is expressed in all L5 neurons, a subset of which are labeled with myr::GFP (green) using FlpOut. L5 neurons are easily identified based on position, as their somas are located directly distal to the lamina plexus defined by R1-R6 axons (blue, mAB24B10), which appears as a thick band. (B and C) Confocal images of L5 growth cones (green) within the outer medulla at 24 hr APF. R7 and R8 growth cones (blue, mAB24B10) define the boundaries of the outer medulla. (D) Quantification of L5 mis-expression experiments at 24 hr APF. N = the number of cells counted. (E–G) Confocal images of L5 axons (green) within the outer medulla in adult flies. Outer medulla layers are identified using R8 and R7 axons (mAB24B10, blue) as a reference. (H) Quantification of L5 mis-expression experiments in adult flies. N = the number of cells counted. (I–O) Mis-expression of dFezf-HA in L2 neurons in MARCM experiments. DFezf-HA was expressed in lamina neuron clones (green, myr::GFP: α-GFP) by 11–164 GAL4 (*Nern et al., 2008*). L2 neurons (magenta) were labeled with myr::tdTomato (α-DsRed) driven by 16H03-LexA. L2 MARCM clones are double positive for myr::GFP and myr::tdTomato and appear white. (I–I''') DFezf-HA (blue, α-HA) is expressed in L2 MARCM clones (white) that are double positive for myr::GFP and myr::tdTomato. (J and K) Confocal images showing L2 growth cone targeting within the outer medulla at 24 hr APF. R7 and R8 growth cones (blue, mAB24B10) define the boundaries of the outer medulla. Scale bar = 5 microns. (L) Quantification of L2 mis-expression experiments at 24 hr APF. N = the number of cells counted. (M and N) Confocal images showing L2 axon targeting within the outer medulla in adult flies. R7 and R8 axons (blue, mAB24B10) define outer medulla layers. Scale bar = 5 microns. (O) Quantification of L2 mis-expression experiments in adult flies. N = the number of cells counted.

DOI: https://doi.org/10.7554/eLife.33962.009

The following source data and figure supplement are available for figure 4:

**Source data 1.** Input data for bar graphs *Figure 4D, H, L, and O*.
DOI: https://doi.org/10.7554/eLife.33962.011
**Figure supplement 1.** DFezf mis-expression in L5 and L2 neurons.
DOI: https://doi.org/10.7554/eLife.33962.010

neurons expressed dFezf-HA, as determined through HA immunostaining (*Figure 4A*, and *Figure 4—figure supplement 1A*), and a subset of L5 neurons were specifically labeled by myr::GFP (*Figure 4A*). We found that the expression of dFezf-HA had no effect on L5 growth cone position in early pupal development. The growth cones of dFezf-expressing L5 neurons innervated the proximal domain of the outer medulla in a manner indistinguishable from wild type neurons (*Figure 4B–D*). In adult flies, L5 terminals are confined to the M5 layer, and L5 neurons also arborize within distal layers M1 and M2 through branching that occurs off the main neurite during mid-late pupal development (*Figure 4E*). DFezf expression did not affect the layer position or morphology of L5 terminals, but caused defects in the distal arborizations of the vast majority of L5 neurons (*Figure 4F–H*). 82% of L5 neurons mis-expressing dFezf displayed reduced distal arborizations that failed to extend into the M2 layer (*Figure 4F,H*), and 15% of neurons extended distal branches into neighboring columns (*Figure 4G,H*). These findings, along with the fact that *dFezf* null L3 neurons inappropriately arborize in distal medulla layers, suggests that dFezf function may antagonize innervation of the distal outer medulla.

To assess the effects of mis-expressing *dFezf* in L2 neurons, we expressed cDNA encoding dFezf-HA in lamina neurons using a pan-lamina neuron GAL4 driver (11–164 GAL4) in MARCM experiments, and labeled L2 neurons using an L2-specific LexA driver (16H03-LexA). The expression of dFezf-HA was assessed through HA immunostaining (*Figure 4I–I′′′*). In these experiments, GAL80 prevented GAL4 expression (and therefore dFezf-HA expression) in lamina neurons that did not undergo mitotic recombination (non-clones), which represented the large majority of lamina neurons. In lamina neuron clones GAL4 induced the expression of dFezf-HA and GFP, and the GFP positive clones that also expressed the L2 marker were confirmed as L2 clones. Only isolated L2 clones within each column were considered, eliminating the possibility of non-autonomous effects from dFezf mis-expression in other lamina neurons in the same column. Using this criteria we found that, in early pupal development, the growth cones of all L2 neurons mis-expressing dFezf-HA inappropriately terminated within the proximal domain of the outer medulla (*Figure 4J–L*), and in adult flies the large majority of these neurons (88%) incorrectly innervated the M3 layer (*Figure 4M–O*). Similar results were found in MARCM experiments wherein a sparser distribution of lamina neuron clones was generated, and only dFezf-HA expressing L2 neurons that were isolated in the home column *and* not next to lamina neuron clones in neighboring columns were considered (*Figure 4—figure supplement 1B–D*). In all cases dFezf-HA expressing L2 neurons still displayed terminal-like swellings within the M2 layer, indicating a partial rather than complete change in layer specificity. These experiments demonstrate that mis-expressing *dFezf* in L2 neurons is sufficient to induce innervation of the proximal domain of the outer medulla, and cause ectopic innervation of the L3 target layer M3. Collectively, our mis-expression studies in L2 and L5 neurons support that dFezf acts instructively to direct growth cone targeting. We hypothesize that dFezf achieves this by antagonizing innervation of the distal outer medulla and promoting innervation of the proximal outer medulla (see discussion).

## DFezf regulates a complex program of cell surface gene expression

To gain insight into the mechanism by which dFezf regulates L3 growth cone targeting, we sought to identify genes that are differentially expressed between wild type and *dFezf* null L3 neurons, as these would represent candidate direct or indirect dFezf targets. To accomplish this, we isolated fluorescently labeled wild type or *dFezf* null L3 MARCM clones by FACS at 40 hr APF (*Figure 5A,B*), prepared cDNA libraries, and sequenced the libraries using an Illumina platform. Five replicates were performed for wild type and mutant genotypes, and to minimize technical variability all ten cDNA libraries were prepared and sequenced at the same time (*Supplementary file 1* contains normalized counts for all samples and genes). Quality control analyses show that the RNA-seq data are of a high caliber and that differences between samples are primarily attributed to genotype (*Supplementary file 2*). Differential expression analysis identified 455 differentially expressed (DE) genes (padj <0.05) (*Figure 5C*, and *Supplementary file 3*), and changes in the expression of several of these were verified at the protein level (*Figure 5—figure supplement 1*). Using previously curated databases (*Kurusu et al., 2008*; *Tan et al., 2015*), we found that 110 of the 455 DE genes are cell surface and secreted genes (*Figure 5—figure supplement 2*), consistent with a role for dFezf in regulating cell-cell communication. Large fractions of DE genes were up or downregulated (190 and 265 genes, respectively) (see *Figure 5C* and *Supplementary file 2*), indicating that dFezf

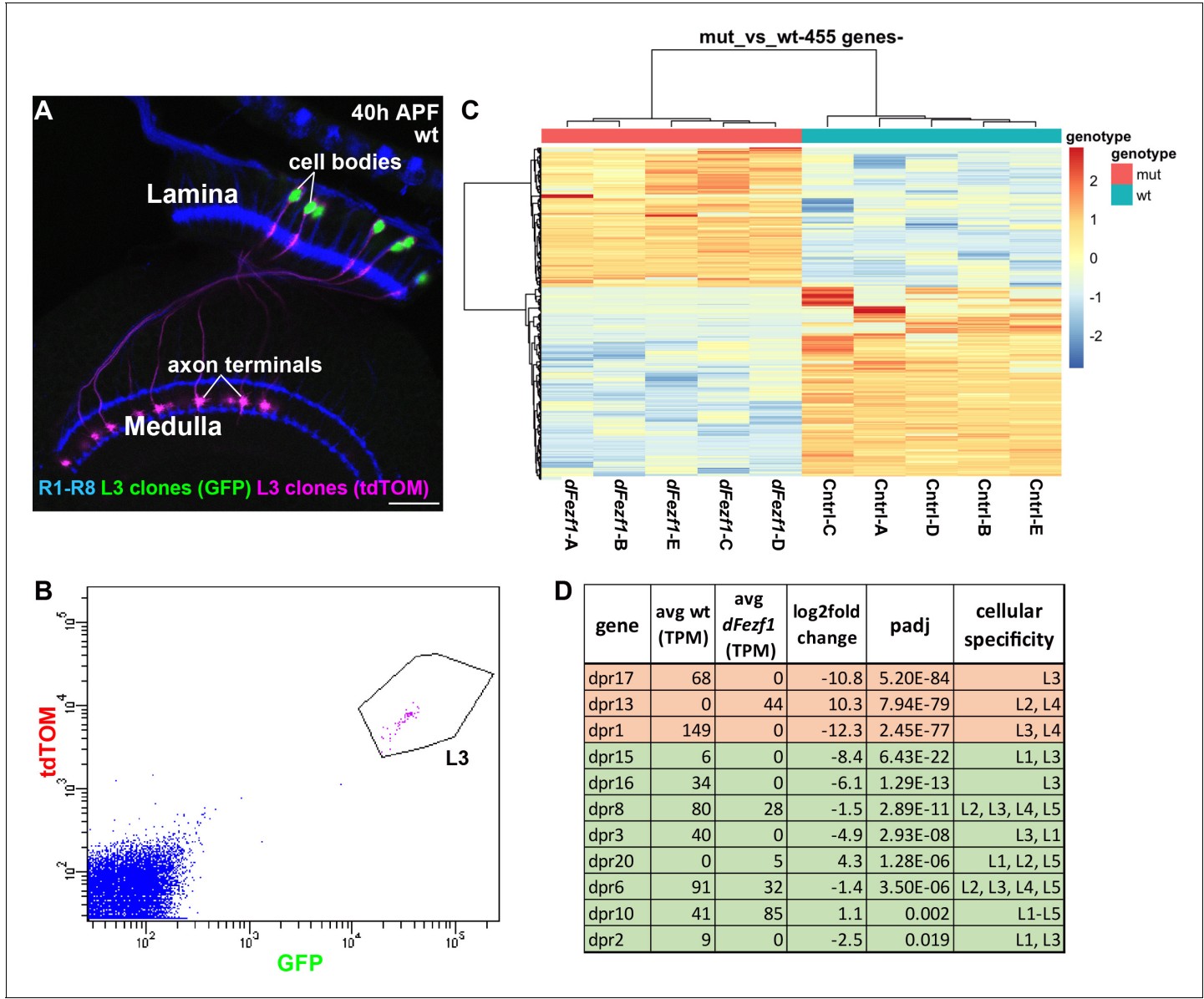

**Figure 5.** DFezf regulates a complex program of cell surface gene expression. (**A and B**) MARCM was performed to generate and specifically label single wild type or *dFezf¹* L3 neurons at 40 hr APF. L3 clones were labeled with a nuclear reporter (H2A-GFP) and a transmembrane reporter (myr:: tdTomato) by 9–9 GAL4. (**A**) A representative confocal image shows that L3 neurons are the only cell type labeled by both GFP (green) and tdTOM (magenta). Photoreceptor neurons (R1-R8) (blue, mAB24B10) are labeled as a reference for the lamina and medulla. Scale bar = 20 microns. (**B**) A representative FACS plot shows that L3 neurons expressing H2A-GFP and myr::tdTomato separated as a tight cluster from background cells (100K events are shown). (**C**) A heat map (row scaled) of differentially expressed genes that includes data from all samples. Each row represents the normalized expression of a differentially expressed gene, and each column represents a sample. Presenting the data in this manner shows that samples clearly separate based on genotype, and that large fractions of differentially expressed genes are either up or downregulated. (**D**) Table showing *dpr* genes that are differentially expressed between wild type and *dFezf* null L3 neurons. The genes are listed in order of significance (padj). The top three most significant DE genes overall, *dprs 17*, *13*, and *1* are shown in red, and the rest of the genes are shown in green. TPM (transcripts per million mapped reads) values are averages across all samples for a given genotype.

DOI: https://doi.org/10.7554/eLife.33962.012

The following figure supplements are available for figure 5:

**Figure supplement 1.** Validation of mRNA changes identified through RNA-seq at the protein level.
DOI: https://doi.org/10.7554/eLife.33962.013

**Figure supplement 2.** Biological categories of DE genes.
DOI: https://doi.org/10.7554/eLife.33962.014

**Figure supplement 3.** Expression of CadN, Sema-1a and Netrin-A/B in wild type and *dFezf* null L3 neurons.

*Figure 5 continued on next page*

*Figure 5 continued*

DOI: https://doi.org/10.7554/eLife.33962.015

function serves to both activate and repress gene expression, either directly or through intermediate regulators. This also suggested that defects in growth cone targeting when *dFezf* is disrupted could result from the downregulation of particular genes, the upregulation of genes, or both.

Since CadN and Sema-1a play a crucial role in L3 growth cone targeting (*Nern et al., 2008*; *Pecot et al., 2013*), we hypothesized that dFezf might regulate their expression as part of a gene program controlling growth cone position. However, *CadN* and *Sema-1a* mRNA levels in wild type and *dFezf* mutant L3 neurons were not significantly different (*Figure 5—figure supplement 3A,B*), demonstrating that dFezf is not necessary for their expression at 40 hr APF. In line with this, *CadN* and *Sema-1a* double mutant L3 neurons display defects in dendrite morphology (*Figure 5—figure supplement 3C*), but dendrites were qualitatively normal in the absence of dFezf function (*Figure 5—figure supplement 3C*), indicating that disrupting *dFezf* does not affect CadN and Sema-1a function. Together, these data strongly support that dFezf acts in parallel to CadN and Sema-1a to regulate L3 growth cone targeting within the outer medulla (see discussion).

Many genes are differentially expressed between wild type and *dFezf* null L3 neurons, and are thus candidates for regulating growth cone targeting. However, ranking the DE genes based on significance revealed several intriguing candidates. For example, using this criteria, the top three genes are *dprs 17*, *13*, and *1*, respectively, which are members of the dpr cell surface family (*Nakamura et al., 2002*). The mRNA levels of all three genes change more than 1000 fold when *dFezf* is disrupted in L3 neurons (*Figure 5D*). Dpr 17 is exclusively expressed in L3 neurons in the lamina and *dpr1* is expressed in L3 and L4 neurons (*Tan et al., 2015*). Both are downregulated in the absence of dFezf function, and for dpr17 we confirmed this at the protein level (*Figure 5—figure supplement 1A–B'*). *Dpr13* is highly expressed in L2 neurons and moderately expressed in L4 neurons (*Tan et al., 2015*), and is upregulated in *dFezf* null L3 neurons. In addition to these, eight other *dpr* genes are differentially expressed between wild type and *dFezf* null L3 neurons (*Figure 5D*), suggesting that *dpr* genes are prominent direct or indirect dFezf targets.

Dpr proteins are immunoglobulin (Ig) domain containing cell surface molecules (21 dprs) that bind heterophilically to dpr interacting proteins (DIPs) (9 DIPs) (*Carrillo et al., 2015*; *Özkan et al., 2013*), which are also cell surface molecules and members of the Ig superfamily (IgSF). Dpr and DIP proteins are expressed in a matching manner between lamina neurons (including L3) and their synaptic targets in the medulla, and specific dpr-DIP interactions, or combinations of them, have been proposed to encode synaptic specificity (*Tan et al., 2015*). Thus, one possibility is that dFezf regulates L3 growth cone position by controlling the expression of specific *dpr* genes, whose products mediate interactions with appropriate DIP-expressing target cells in the medulla. In the absence of dFezf function, an abnormal complement of *dprs* is expressed (L3 *dprs* are downregulated and *dprs* expressed by other lamina neurons are upregulated), and this may promote interactions with incorrect target neurons, altering growth cone position and laminar specificity (see discussion).

In summary, a large number of genes change in their levels of expression when *dFezf* is disrupted in L3 neurons, and similar numbers of these are up or downregulated. Nearly a quarter of the DE genes are cell surface and secreted molecules, suggesting that dFezf regulates communication between L3 and other cells. *Dpr* genes are strong candidates for acting downstream of dFezf to regulate growth cone targeting. However, as many candidates exist, additional studies are necessary to identify the genes relevant for this process (see discussion).

## DFezf is necessary and sufficient for netrin expression

L3-derived Netrin is necessary for the M3-specific innervation of R8 photoreceptor axons (*Pecot et al., 2014*; *Timofeev et al., 2012*). Our RNA-seq analyses revealed that the mRNA levels of *Netrin-A* and *B (NetA/B)* were drastically reduced in *dFezf* null L3 neurons compared to wild type neurons at 40 hr APF (*Figure 5—figure supplement 3A,B*). This suggested that, in addition to cell autonomously regulating L3 targeting, dFezf non-autonomously regulates R8 layer specificity via activation of *Netrin* expression in L3 neurons. To test this we first assessed Netrin protein expression in L3 neurons during development. Previous gene expression and genetic studies implied that L3

neurons express Netrin protein (*Pecot et al., 2014*; *Tan et al., 2015*; *Timofeev et al., 2012*), but this had not been shown directly. To address this we used a NetB-GFP protein trap, wherein endogenous NetB is tagged with GFP. We focused on NetB protein because *NetB* is expressed at much higher levels than *NetA* in L3 neurons (*Figure 5—figure supplement 3A,B*) (*Tan et al., 2015*), and the expression of *NetB* in L3 neurons is sufficient to support proper R8 layer specificity in the background of a *NetA/B* deletion (*Timofeev et al., 2012*). At 24 hr APF we observed NetB expression in the perinuclear region of the oldest lamina neurons (most anterior) (*Figure 6A–D'*). Using markers specific for different lamina neuron subclasses we discovered that NetB is expressed by L3, L4 and L5 neurons, but not L1 or L2 neurons. We assessed NetB expression in L3 by labeling L3 somata with FLAG-tagged dFezf (dFezf-FLAG) expressed from its native promoter within a BAC (*Figure 6—figure supplement 1*). At 48 hr APF NetB expression remained specific to L3, L4 and L5 neurons in the lamina, although it was most strongly expressed in L3 neurons at this stage (*Figure 6E–H*). Thus, while NetB immunoreactivity is first detected within the M3 layer at ~40 hr APF (*Timofeev et al., 2012*), which coincides with the segregation of L3 growth cones into the developing M3 layer, L3 neurons express NetB much earlier in development. In addition, we found that NetB is also expressed by L4 and L5 neurons during pupal development, and thus may regulate circuit assembly within multiple developing layers.

To test whether dFezf is required for NetB protein expression we examined NetB expression during pupal development when dFezf function was disrupted in L3 neurons. Disrupting *dFezf* in single L3 neurons in MARCM experiments drastically reduced NetB immunostaining in L3 cell bodies at 48 hr APF (*Figure 7A–B'*). To assess whether *dFezf* is necessary for NetB expression in the M3 layer, wherein it is thought to be secreted from L3 growth cones, we performed RNAi experiments to disrupt *dFezf* in the vast majority of L3 neurons. The expression of *dFezf* RNAi in the lamina eliminated detectable dFezf immunoreactivity in L3 neurons (*Figure 7C,D*), and considerably reduced NetB immunostaining in L3 somas and the developing M3 layer at 48 hr APF (*Figure 7E,F*). Together, these experiments demonstrate that *dFezf* is cell autonomously required for NetB protein expression in L3 neurons.

To determine if dFezf is sufficient to induce NetB expression, we mis-expressed dFezf-HA in lamina neurons using a pan-lamina GAL4 driver and assessed NetB expression in L2 neurons (*Figure 7G–J*). L2 neurons mis-expressing *dFezf* still expressed Bab1 (*Figure 7J*), a transcription factor shown to be selectively expressed in L2 neurons in the lamina (*Tan et al., 2015*), indicating that *dFezf* expression did not cause an obvious change in cell fate. We found that, while no wild type L2 neurons expressed NetB at 48 hr APF (*Figure 7I*), 99% of dFezf-HA expressing L2 neurons ectopically expressed NetB (*Figure 7J*). Thus, in the context of L2 neurons dFezf is sufficient to induce NetB expression. Collectively, our RNA-seq and protein expression experiments support that dFezf, either directly or through intermediate regulators, activates *Netrin* expression in L3 neurons.

## DFezf in L3 neurons is non-autonomously required for R8 layer specificity

To directly determine if dFezf function in L3 neurons is required for R8 layer specificity, as is predicted by its role in regulating *Netrin* expression, we disrupted *dFezf* via RNAi and assessed R8 axon targeting using an R8-specific marker. Normally, R8 axons terminate in the M3 layer (*Figure 7K,M*). When *dFezf* was disrupted 45% of R8 axons terminated within inappropriate layers (*Figure 7L,M*). 29% of the axons terminated in layers proximal to M3 (i.e. M4-M6) and 16% terminated in distal layers (i.e. M0-M2). The penetrance of mis-targeting in our experiments is similar to that reported by Timofeev and colleagues in a *Netrin* null background (53%) (*Timofeev et al., 2012*). However, they reported that only 2% of R8 axons mis-targeted to more proximal layers (versus 29% in our experiments). Differences in R8 mis-targeting between ours and previous experiments could reflect nonspecific effects of disrupting dFezf in L3 neurons (e.g. due to mis-targeting L3 axons), or indicate that L3 neurons provide a dFezf-dependent cue in addition to Netrin that is important for the targeting of R8 axons (see discussion). Nonetheless, our experiments show that *dFezf* in L3 neurons is required for R8 layer specificity. Together with previous research, these findings along with our analyses of *Netrin* gene and protein (NetB) expression indicate that dFezf regulates R8 layer specificity, to a large extent, through activation of *Netrin* expression in L3 neurons.

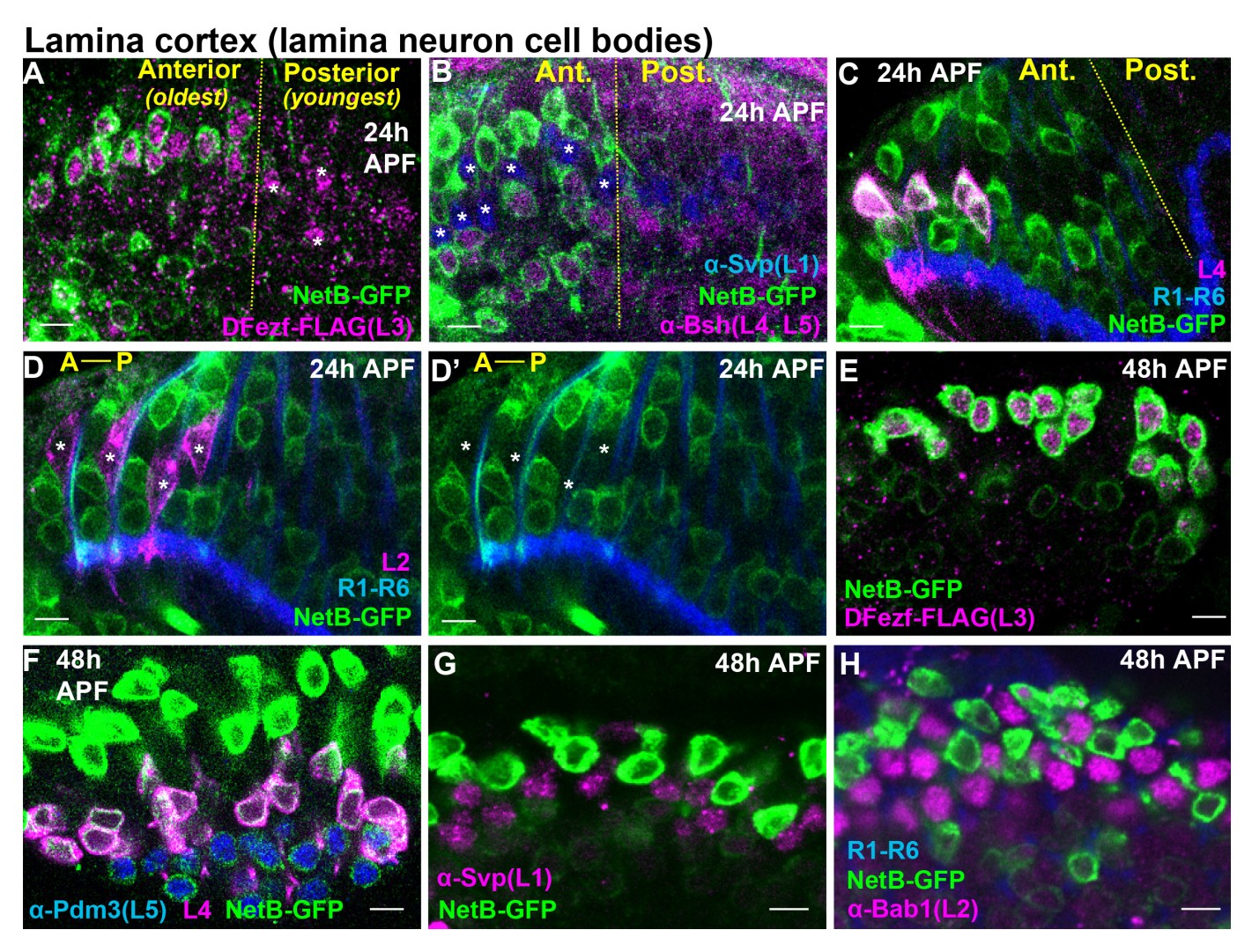

**Figure 6.** Cellular expression of NetB in the lamina during pupal development. (A–H) NetB expression in the lamina at 24 or 48 hr APF. Endogenous NetB was visualized using a NetB-GFP protein trap (α-GFP). All confocal images are representative examples, and at least five brains were examined per genotype. All scale bars = 5 microns. (A–D') In the lamina, lamina neurons are born in a wave (Anterior = oldest, Posterior = youngest). The yellow dashed lines in (A–C) indicate the anterior-posterior region of the lamina where NetB expression is first observed. In D and D' anterior-posterior is indicated by A-P. (A) At 24 hr APF, NetB (green) was expressed in the anterior side of the developing lamina (to the left of the yellow dashed line) containing the oldest lamina neurons. NetB was expressed in L3 neurons (magenta, dFezf-FLAG). The youngest L3 neurons on the posterior side of the lamina that had not turned on NetB expression are indicated by asterisks. (B) At 24 hr APF, NetB (green) was detected in L4 and L5 neurons (magenta, α-bsh) in the oldest region of the lamina (to the left of the yellow dashed line). NetB (green) was not detected in L1 neurons (blue, α-svp, asterisks in the anterior region). (C) Consistent with findings shown in B, at 24 hr APF NetB (green) was expressed in the oldest L4 neurons (magenta) in the anterior region of the lamina (left of the dashed yellow line). L4 neurons were labeled using an L4-specific driver (31C06-LexA, LexAop-myr::tdTomato). At this stage 31C06-LexA drives expression in a small population of L4 neurons in the anterior region of the lamina. R1-R6 axons (blue. mAB24B10) provided a reference for the lamina. The image shown here is taken from a pupa that is slightly older than in A and B, as NetB is expressed in nearly all the lamina neurons in the field of view. (D and D') NetB (green) was not expressed in L2 neurons (magenta) at 24 hr APF. L2 neurons were visualized using a specific driver (16H03-LexA, LexAop-myr::tdTomato) that drives expression in the most anterior (oldest) L2 neurons at this stage. R1-R6 axons (blue, mAB24B10) serve as a reference for the lamina. Asterisks show the positions of L2 somas. The image shown here is taken from a pupa that is slightly older than in A and B, as NetB is expressed in all the lamina neurons in the field of view. (E) NetB (green) is expressed in all L3 neurons (magenta, dFezf-FLAG) at 48 hr APF. (F) All L4 (magenta, 31C06-LexA, LexAop-myr::tdTomato) and L5 (blue, α-Pdm3) neurons express NetB at 48 hr APF. NetB expression in L4 and L5 neurons is weaker than in L3 neurons (green only). (G) NetB (green) is not expressed in L1 neurons (magenta, α-Svp) at 48 hr APF. (H) NetB (green) is not expressed in L2 neurons (magenta, α-Bab1) at 48 hr APF.

DOI: https://doi.org/10.7554/eLife.33962.016

The following figure supplement is available for figure 6:

*Figure 6 continued on next page*

*Figure 6 continued*

**Figure supplement 1.** Characterization of dFezf-FLAG (BAC).
DOI: https://doi.org/10.7554/eLife.33962.017

## Disrupting dFezf in L3 neurons does not affect the layer innervation of Tm9 neurons

Within the M3 layer L3 axons synapse onto the dendrites of transmedullary nine neurons (Tm9) (*Gao et al., 2008*; *Takemura et al., 2013*; *Takemura et al., 2015*), and synapses between R8 and Tm9 have also been reported (*Gao et al., 2008*). Tm9 neurons are a subtype of projection neurons that transmit information from the medulla to the lobula neuropil. As disrupting *dFezf* in L3 neurons caused defects in L3 and R8 layer innervation but did not impair the formation of presynaptic sites in L3 neurons (*Figure 2K,L*), we assessed whether connectivity with Tm9 neurons is perturbed under these conditions. To test this we used cell-specific drivers to assess overlap between L3 axons and Tm9 dendrites within the same medulla columns during development, under conditions when L3 neurons were wild type or mutant for *dFezf* in MARCM experiments. During early pupal development (~24 hr APF) in wild type columns, L3 growth cones and Tm9 dendrites targeted to similar depths within the outer medulla, but appeared to be restricted to opposite sides of the column (*Figure 8A,B–B"*). In contrast, by mid-pupal development (~48 hr APF) L3 growth cones and Tm9 dendrites overlapped significantly within the developing M3 layer (*Figure 8C,D–D"*). In adult flies L3 terminals were flattened and overlapped with the proximal portion of Tm9 dendrites within the M3 layer (*Figure 8E,F–F"*).

When *dFezf* was disrupted in L3 neurons the positions of L3 growth cones and Tm9 dendrites within the same columns were altered in early pupal development (*Figure 8G–G"*). Frequently, the growth cones and dendrites were positioned at different depths within the outer medulla. At mid-pupal development L3 growth cones and Tm9 dendrites did not overlap within the column, with most of the growth cones innervating developing layers in the distal medulla (M1-M2) while Tm9 dendrites innervated the developing M3 layer normally (*Figure 8H–H"*). In adult flies, we found that compared to wild type L3 clones, *dFezf*[1] clones were weakly labeled by an L3-specific GAL4 driver (9–9 GAL4) (*Figure 8I,I"*). Despite the weak labeling, we observed that L3 axon terminals and Tm9 dendrites innervated different depths within the neuropil, and the morphologies of Tm9 dendrites in columns containing wild type or *dFezf*[1] L3 neurons were indistinguishable (*Figure 8I–I"*). The decreased labeling observed in mutant clones is likely due to a reduction in 9–9 GAL4 activity during pupal development rather than cell death, as *dFezf*[1] L3 neurons in adult flies were strongly labeled with a pan-neuronal driver (i.e. the *brp* locus) (see *Figure 2G–I,L,N*). Thus, dFezf may be required to maintain 9–9 GAL4 expression in L3 neurons.

To summarize, in these experiments we did not observe overlap between the axons of *dFezf* null L3 neurons and Tm9 dendrites during development or in adult flies, strongly pointing to disrupted connectivity between these neurons under these conditions. In addition, it is also likely that dFezf is partially required for R8-Tm9 connectivity, as a significant number of R8 axons terminate in layers distal to M3 in the absence of dFezf function (*Figure 7L,M*), and R8 axons and Tm9 dendrites are unlikely to overlap in these columns.

## Discussion

Laminar organization of synaptic connections is a principal feature of nervous system structure. Thus, determining how cells arrange their connections into layered networks is critical for understanding how the nervous system is assembled. Molecules that are necessary for layer specificity in particular neuron types have been identified in different systems. However, molecular strategies that organize the assembly of specific layers have remained elusive. Here, we illuminate a transcriptional mechanism that coordinates different cell types to the same layer in the *Drosophila* visual system (summarized in *Figure 9*). Based on our findings and previous studies, we propose that specific layers are built in a stepwise manner through use of transcriptional modules that cell-intrinsically instruct targeting to specific layers, and cell-extrinsically recruit other circuit components. Moreover, based on

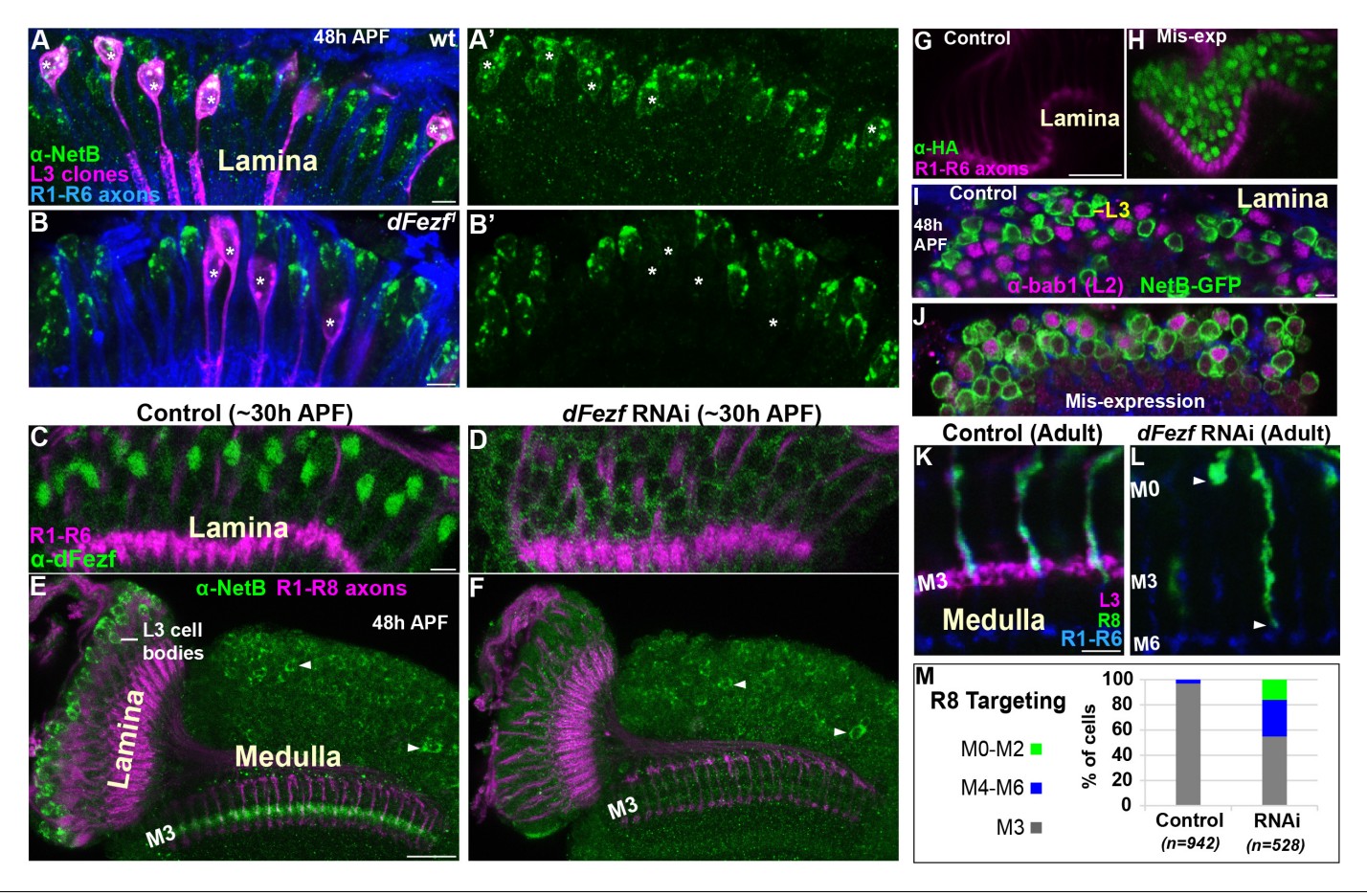

**Figure 7.** DFezf non-autonomously regulates R8 layer specificity through activation of *Netrin* expression in L3 neurons. (**A–B'**) NetB expression (green, α-NetB) in the somas of wild type or *dFezf¹* L3 neurons was assessed at 48 hr APF in the lamina in MARCM experiments using confocal microscopy. L3 clones (magenta) expressed myr::GFP (9–9 GAL4) and were visualized using a GFP antibody. (**A–A'**) Asterisks indicate the positions of wild type L3 clones (magenta). The images are representative of NetB expression (green) assessed in five brains. (**A**) R1-R6 axons (blue, mAB24B10) were used as a reference for the lamina. Scale bar = 5 microns. (**B–B'**) Asterisks indicate the positions of *dFezf¹* L3 clones (magenta). The images are representative of NetB expression (green) assessed in six brains. (**B**) R1-R6 axons (blue, mAB24B10) were used as a reference for the lamina. Scale bar = 5 microns. (**C and D**) *DFezf* RNAi was expressed in lamina neurons and their precursor cells using 9B08-GAL4. DFezf expression (green) in L3 neurons was assessed through immunostaining (α-dFezf). R1-R6 axons (magenta, mAB24B10) provided a reference for the lamina. (**C**) Representative confocal image of dFezf expression in control flies, which include flies containing *dFezf* RNAi only or 9B08-GAL4 (seven brains total). Scale bar = 5 microns. (**D**) Shows a representative confocal image of dFezf expression in knockdown flies (eight brains examined). (**E and F**) NetB (green, α-Net-B) expression in the lamina and within the M3 layer was assessed in control brains and when *dFezf* RNAi was expressed in lamina neurons and their precursors using 9B08-GAL4. R1-R8 axons (magenta, mAB24B10) provided a reference for the lamina and outer medulla. Arrowheads indicate NetB expression in medulla neurons. (**E**) Shows a representative confocal image of NetB expression (green) in control flies, which contain *dFezf* RNAi only or 9B08-GAL4 (four brains total). Net-B was prominently observed in L3 cell bodies in the lamina, within the M3 layer in the medulla and in medulla neuron cell bodies (arrowheads) on a consistent basis. Scale bar = 20 microns. (**F**) A representative confocal image from flies containing *dFezf* RNAi +9B08 GAL4 (six brains). NetB (green) was consistently reduced in L3 cell bodies in the lamina and the M3 layer, but not in medulla neuron cell bodies (arrowheads). (**G–J**) DFezf-HA was expressed broadly in lamina neurons by 11–164 GAL4. (**G and H**) Confocal images of dFezf-HA expression (green, α-HA) in lamina neurons (~24 hr APF). The lamina was identified by visualizing R1-R6 axons (magenta, mAB24B10). (**G**) Representative confocal image from flies containing DFezf-HA or 11–164 GAL4 only (at least five brains). No HA labeling was detected in the lamina of these flies. (**H**) Shows a representative confocal image from flies containing DFezf-HA +11–164 GAL4 (at least five brains). Prominent HA staining (green) was always observed in the lamina of these flies at 24 hr APF. (**I and J**) Confocal images of NetB expression (green) (NetB-GFP, α-GFP) in the lamina at 48 hr APF. L2 neurons (magenta) were labeled using α-bab1, and R1-R6 axons (blue, mAB24B10) provided a reference for the lamina. (**I**) Representative confocal image from flies containing DFezf-HA or 11–164 GAL4 (four brains). No L2 neurons (magenta) expressed Net-B (green) (n = 308). NetB-expressing cells in the image are L3 neurons. Scale bar = 5 microns. (**J**) Representative confocal image from flies containing DFezf-HA +11–164 GAL4 (four brains). 99% of L2 neurons (magenta) expressed NetB (green) (n = 310). (**K–M**) *DFezf* RNAi was expressed in lamina neurons and their precursors using 9B08-GAL4 and R8 axon targeting was assessed within the outer medulla. (**K and L**) Confocal images showing R8 axon targeting within the outer medulla in adult flies. R8 axons (green) were visualized using an R8-specific marker (Rh6-GFP, α-GFP). L3 axons (magenta) are labeled by 22E09-LexA, LexAop-myr::tdTomato. *DFezf* knockdown inhibits 22E09-LexA

*Figure 7 continued on next page*

*Figure 7 continued*

activity, and serves as a positive control for the efficacy of *dFezf* RNAi. R7 and R8 axons (blue, mAB24B10) provide a reference for outer medulla layers. (K) A representative image from flies containing *DFezf* RNAi or 9B08-GAL4 only (six brains total). Scale bar = 5 microns. (L) A representative image from flies containing *DFezf* RNAi +9B08 GAL4 (four brains). Arrowheads indicate the depths at which R8 axons terminate. (M) Quantification of R8 axon targeting. N = the number of R8 axons counted.

DOI: https://doi.org/10.7554/eLife.33962.018

The following source data is available for figure 7:

**Source data 1.** Input data for bar graph *Figure 7M*.

DOI: https://doi.org/10.7554/eLife.33962.019

research in the mammalian nervous system we propose that this represents an evolutionarily conserved strategy for building laminar-specific neural circuits (see below).

## DFezf cell-intrinsically instructs axon target specificity

Early in pupal development, lamina neuron axons project into the outer medulla and terminate in two broad domains (*Figure 1C*), which later give rise to refined layers (*Figure 1B*). These broad domains represent a primitive form of layer organization, and thus determining how they are established is likely to be central to understanding how specific layers are constructed. The targets of lamina neuron axons within broad domains are unknown, and could include synaptic partners, intermediate targets, guidepost cells, and molecules embedded in the extracellular matrix. Our findings demonstrate that dFezf regulates broad domain specificity in L3 neurons, and indicate that this step in development is crucial for L3 laminar specificity, synaptic connectivity, and for the proper assembly of the M3 layer.

We show that dFezf functions in parallel to CadN and Sema-1a in L3 neurons to regulate targeting to the proximal domain of the outer medulla. We envision three models by which proper growth cone position could be achieved through the combined actions of these molecules. (1) In the first model precise growth cone position is achieved by preventing innervation of inappropriate regions. Here, dFezf prevents growth cone termination within the distal domain of the outer medulla, potentially by repressing the expression of cell surface molecules that interact with targets in this region. And CadN and Sema-1a prevent the growth cones from extending beyond the proximal domain, into the serpentine layer and inner medulla. Thus, proper growth targeting is achieved through mechanisms that prevent both superficial and deeper innervation. (2) Alternatively, growth cone position may be controlled though a direct targeting mechanism. In this model, dFezf could promote targeting to the proximal domain of the outer medulla by activating the expression of cell surface genes that interact with targets in this region. In this context, CadN and Sema-1a would then regulate growth cone consolidation within the proximal domain of the outer medulla. (3) And finally, dFezf function could serve to both prevent termination in the distal domain of the outer medulla, and direct targeting to the proximal domain (combination of (1) and (2)).

In the absence of dFezf function, cell surface genes that are normally expressed by L3 neurons are downregulated, and cell surface genes expressed by other lamina neurons become upregulated. Thus, we favor the third model involving both prevention of superficial termination and directed growth cone targeting. We hypothesize that dFezf function serves to (1) activate the expression of cell surface genes that promote interactions with targets in the proximal domain of the outer medulla, and (2) repress the expression of cell surface genes that mediate interactions with targets in the distal domain. DFezf could accomplish this by directly binding the loci of cell surface genes to control their expression, by acting through intermediate factors, or a combination of both. *Dpr* genes are prominent dFezf targets, and one interesting possibility is that dFezf regulates growth cone targeting by controlling a program of *dpr* gene expression. Dprs that are activated by dFezf function in L3 neurons may mediate interactions with DIP-expressing synaptic targets in the proximal domain of the outer medulla. In contrast, dprs that become upregulated in *dFezf* mutant L3 neurons may mediate inappropriate interactions with DIP-expressing targets in the distal domain of the outer medulla.

Future studies will be dedicated to determining which dFezf target genes regulate L3 growth cone targeting, and the mechanisms by which dFezf regulates their expression. Since, in addition to

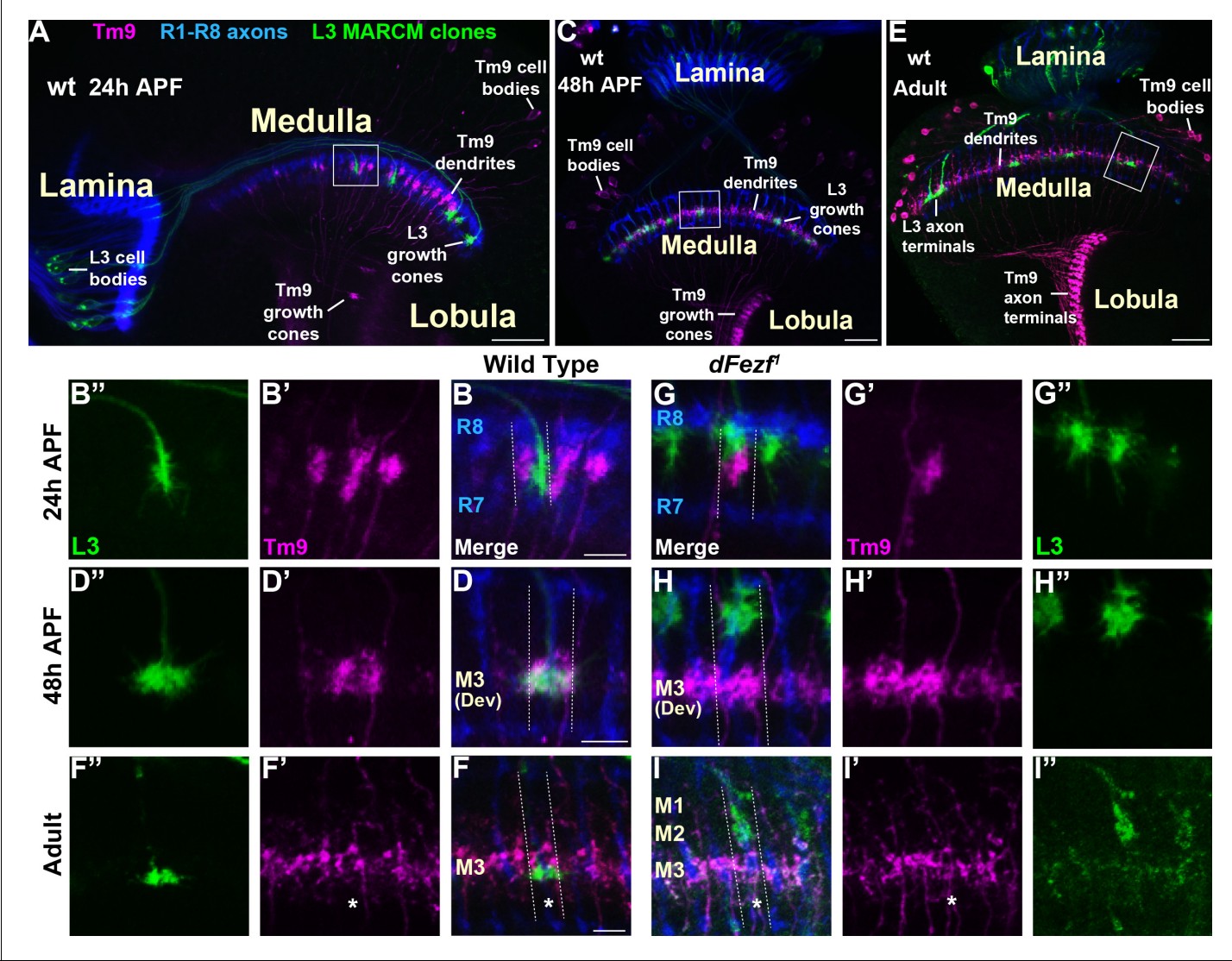

**Figure 8.** Disrupting dFezf in L3 neurons does not affect the layer innervation of Tm9 neurons. (A–I") Confocal images from MARCM experiments analyzed at 24 hr APF, 48 hr APF or in newly eclosed adults. The images shown are representative of neurons assessed from at least five brains per genotype per time-point. L3 clones (green) expressed myr::GFP (9–9 GAL4) and were visualized using a GFP antibody, wild type Tm9 neurons (magenta) expressed myr::tdTomato (24C08-LexA) and were visualized with a DsRed antibody. R1-R8 axons (blue, mAB24B10) define the lamina and outer medulla. (A, C, E) Broad views showing the entire lamina and medulla and the morphologies of wild type L3 (green) and Tm9 neurons (magenta). The solid boxed regions show examples of the regions of the outer medulla shown in B-B', D-D', F-F', G-G', H-H', and I-I'. (B, D, F, G, H, I) Dashed lines indicate the estimated boundaries of a medulla column using R7 and R8 axons (blue) as a reference. (F and F", I and I') Asterisks mark the positions of medulla columns containing L3 clones.

DOI: https://doi.org/10.7554/eLife.33962.020

dprs, many cell surface genes change in their levels of expression in the absence of dFezf function, RNA-seq analyses performed at earlier stages of development and in dFezf mis-expression experiments (e.g. in L2 neurons) may help narrow down the list of relevant target genes. In addition, to circumvent issues of functional redundancy it will be important to undertake combinatorial genetic approaches targeting similar genes (e.g. the dprs) in loss and gain of function experiments. That a number of transcriptional regulators are differentially expressed between wild type and dFezf mutant L3 neurons (*Figure 5—figure supplement 2*) suggests that dFezf, at least in part, acts through intermediate regulators to control gene expression. Identification of direct dFezf targets will provide a way of testing this and addressing how dFezf controls gene expression in L3 neurons.

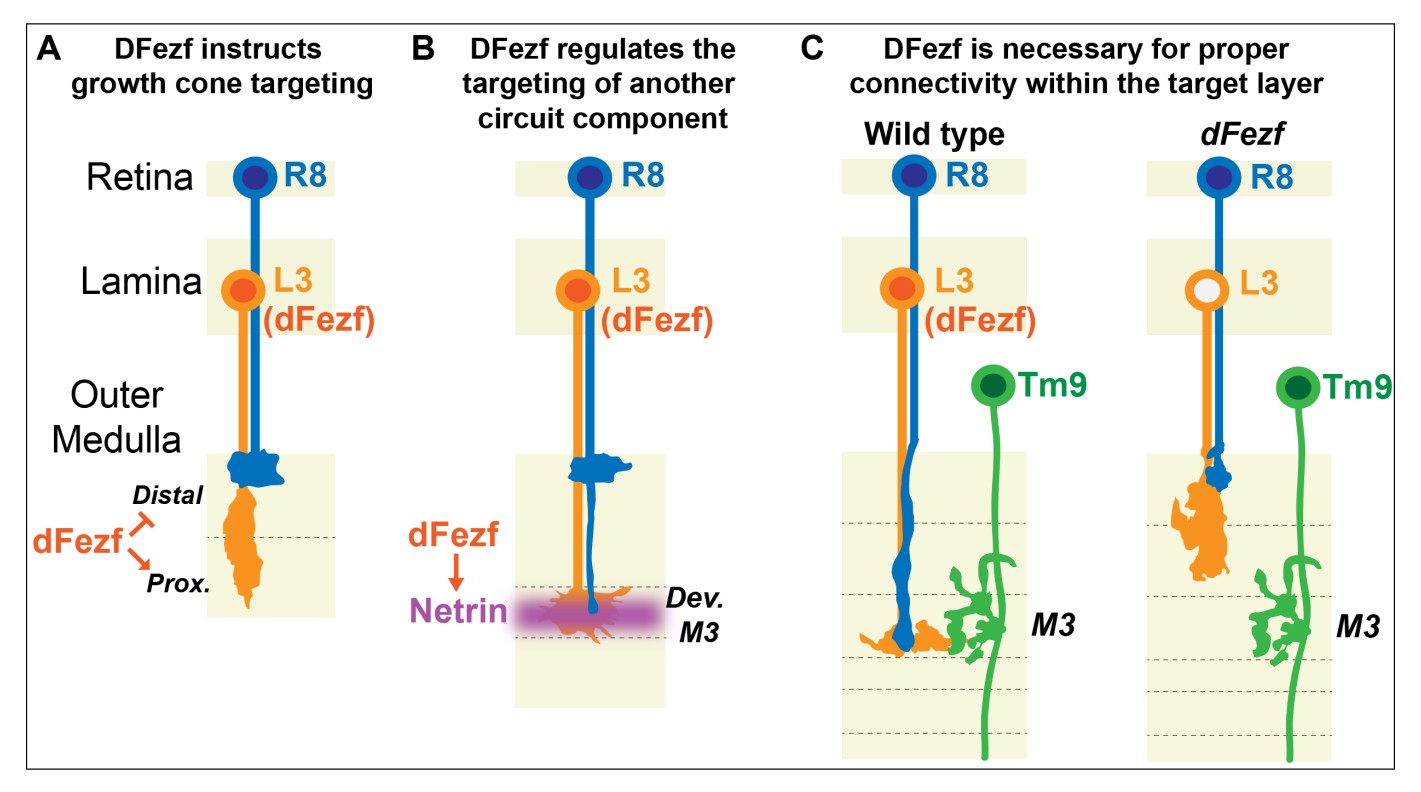

**Figure 9.** DFezf coordinates the formation of laminar-specific connections. (**A**) Early in medulla development, dFezf promotes the targeting of L3 growth cones to the proximal versus distal domain of the outer medulla. DFezf may regulate this step by controlling a program of dpr gene expression. (**B**) L3 growth cones segregate into the developing M3 layer and secrete Netrin, which regulates the attachment of R8 growth cones within the layer. DFezf also regulates this step by activating the expression of *Netrin* in L3 neurons. (**C**) Within the M3 layer, L3 and R8 axons synapse onto Tm9 dendrites. When dFezf function is lost in L3 neurons, L3 and R8 axons innervate inappropriate layers while Tm9 dendrites innervate the M3 layer normally. As a result, connectivity with Tm9 neurons is disrupted.

DOI: https://doi.org/10.7554/eLife.33962.021

## DFezf regulates layer innervation through a cell-extrinsic mechanism

Our genetic investigation of dFezf function in L3 neurons revealed a cell-extrinsic role for dFezf in regulating R8 layer specificity through the activation of *NetA/B*. Whether these genes are direct or indirect dFezf targets remains unclear. Preliminary bioinformatics analyses did not identify dFezf binding motifs within the *NetA/B* loci. However, additional experiments are necessary to rigorously assess this, and determine how *NetA/B* expression is controlled in L3 neurons.

Many different neurons have been shown to form synapses within the M3 layer (*Takemura et al., 2013*; *Takemura et al., 2015*), and whether L3-derived Netrin is necessary for the M3-specific innervation of neurons other than R8 remains an open question. The Netrin receptor Frazzled is broadly expressed in the medulla (*Timofeev et al., 2012*), and Netrin released from L3 growth cones is well-suited to serve as an M3-specific signal to other cell types, due to its concentration within the layer. Genetic studies disrupting dFezf function and assessing the morphologies of neurons that innervate M3 would provide a way of testing whether L3-derived Netrin broadly organizes M3 circuitry, or specifically controls R8 layer innervation.

In the absence of *dFezf* function in L3 we observed R8 axons inappropriately innervating distal layers (i.e. M0-M2), as expected in the absence of Netrin function (*Akin and Zipursky, 2016*; *Timofeev et al., 2012*), and a significant fraction of R8 axons terminating in proximal outer medulla layers (M4-M6), which was unexpected. In vivo time lapse imaging studies showed that the axons of *Frazzled* (Netrin receptor) null R8 neurons targeted to M3 normally and then retracted, indicating that Netrin/Frazzled signaling is required for growth cone stabilization *within* the target layer, but is dispensable for recognition *of* the target layer (*Akin and Zipursky, 2016*). Thus, additional

molecules regulate R8 layer specificity. One interesting possibility is that in addition to *Netrin*, dFezf regulates the expression of a gene(s) important for R8 axons to 'recognize' the M3 layer, and in the absence of this cue a subset of R8 axons overshoot M3 and innervate proximal layers. Consistent with this idea, the genetic ablation of L3 neurons caused a significant percentage of R8 axons to terminate in proximal outer medulla layers (*Pecot et al., 2014*). In addition to *Net-A/B*, dozens of cell surface or secreted genes are downregulated in L3 neurons at 40 hr APF when *dFezf* is disrupted. Thus, there are a number of candidates that could act downstream of dFezf in parallel to Netrin to regulate R8 layer specificity. Disrupting these genes in L3 neurons and visualizing R8 axons would provide a way of identifying additional L3-derived cues necessary for R8 layer innervation.

## A transcriptional mechanism for controlling the stepwise formation of layer-specific circuits

Previous studies indicate that medulla layers are not pre-established regions that serve as a template for circuit formation, but rather that layers are built over time from broad domains as the neurites of different cell types are added in a precise order. Thus, central to elucidating the molecular and cellular logic underlying assembly of the medulla network is identifying the molecular and cellular components involved in the sequence of interactions giving rise to specific layers, and discovering commonalities or connections between interactions and how different layers are assembled. Insight into the cellular mechanisms giving rise to the M3 layer circuitry was provided by the discovery that R8 innervation of M3 relies on a signal from L3 axons (Netrin) (*Pecot et al., 2014*; *Timofeev et al., 2012*), which innervate M3 at an earlier stage despite being born later in development (*Selleck and Steller, 1991*). These studies and the fact that L3 and R8 neurons contribute input to common pathways suggest that the cellular mechanisms governing circuit formation reflect functional relationships between neurons, rather than birth order.

Less progress has been made in identifying the molecular logic underlying the stepwise assembly of layer-specific circuits. This is likely due in part, to the fact that how neurons innervate particular layers is cell type-specific, rather than layer-specific. For example, while R8 neurons depend on Netrin for layer specificity, Netrin is dispensable for the layer innervation of L3 and Tm9 neurons (unpublished results). Thus, identifying commonalities in the mechanisms used by neurons to innervate layers has been challenging. However, here we show that L3 and R8 layer innervation is linked through dFezf, which controls a program of gene expression that regulates the layer innervation of both neurons. Thus, while L3 and R8 utilize different molecules to achieve layer specificity, the expression of key molecules required by each neuron is controlled by the same transcription factor.

Using the same transcriptional pathway or module, to control genes that function cell-intrinsically and cell-extrinsically to regulate layer innervation, represents a simple mechanism for controlling the series of interactions that lead to circuit formation. Employing specific transcriptional modules in specific neurons would ensure that those neurons innervate the correct target layer, and also express the genes necessary for the targeting of other circuit components. We hypothesize that this strategy is widely employed to regulate the assembly of circuitry within the medulla. For example, cell-specific transcription factors analogous to dFezf in L3 have been identified in each of the other lamina neuron subtypes (*Tan et al., 2015*), and the positions of lamina neuron arborizations in the medulla are well suited to produce signals that regulate the precise layer innervation of other cell types (*Figure 1B*). Indeed, CadN-based interactions between L2 terminals and L5 axons are necessary for L5 branching within the M2 layer (*Nern et al., 2008*). Thus, we hypothesize that the function of cell-specific transcriptional modules in lamina neurons coordinates the assembly of discrete outer medulla layers.

We also speculate that the function of dFezf in L3 neurons is conserved in laminated regions of the mammalian nervous system. For instance, in the inner plexiform layer of the mouse retina, which is analogous to the medulla in structure and function (*Sanes and Zipursky, 2010*), Fezf1 is expressed in a subset of bipolar cells that innervate specific sublaminae (i.e. layers) (*Shekhar et al., 2016*). Bipolar cells are analogous to lamina neurons, and we hypothesize that Fezf1 in specific types of bipolar cells functions analogously to dFezf in L3 to coordinate laminar-specific connectivity.

In the cerebral cortex of mice, Fezf2 cell-intrinsically controls the identity of subcortically projecting pyramidal neurons that predominantly reside in layer V (*Chen et al., 2005a*; *Chen et al., 2005b*; *Molyneaux et al., 2005*), and pyramidal neuron identity was shown to be important for the laminar positioning of specific classes of GABAergic neurons (*Lodato et al., 2011*). While it remains

unclear as to whether Fezf2 in pyramidal cells regulates the expression of molecules that act non-autonomously to recruit specific inhibitory neurons (analogous to Netrin), one interesting possibility is that Fezf2 in the cortex organizes laminar-specific circuitry through cell-intrinsic and cell-extrinsic mechanisms similar to dFezf in the medulla. In conclusion, we expect our findings regarding the function of dFezf in the *Drosophila* visual system reflect an evolutionarily shared strategy for assembling laminar-specific neural circuits.

# Materials and methods

**Key resources table**

| Reagent type (species) | Designation | Source or reference | Identifiers | Add. info. |
|---|---|---|---|---|
| Strain (*Drosophila melanogaster*) | 9–9 GAL4 | Gift from U. Heberlein (***Nern et al., 2008***) | N/A | |
| Strain (*D. melanogaster*) | P{10XUAS-IVS-myr::GFP}attP2 | Bloomington Drosophila Stock Center | RRID:BDSC_32197 | |
| Strain (*D. melanogaster*) | P{tubP-GAL80}LL10 P{neoFRT}40A | Bloomington Drosophila Stock Center | RRID:BDSC_5192 | |
| Strain (*D. melanogaster*) | P{R27G05-FLPG5.PEST}attP40 | Bloomington Drosophila Stock Center | RRID:BDSC_55765 | |
| Strain (*D. melanogaster*) | {LexAop-myr::tdTomato} su(Hw)attP5 | S.L. Zipursky (***Chen et al., 2014***) | N/A | |
| Strain (*D. melanogaster*) | P{20XUAS-RSR.PEST}attP2 | Bloomington Drosophila Stock Center | RRID:BDSC_55795 | |
| Strain (*D. melanogaster*) | 79C23S-GS-RSRT-STOP-RSRT-smFP_V5-2A-LexA | this paper | N/A | See details in 'Construction of transgenic animals' session in Materials and methods |
| Strain (*D. melanogaster*) | dFezf2 | C.Y. Lee (***Weng et al., 2010***) | N/A | |
| Strain (*D. melanogaster*) | dFezf1 | C.Y. Lee (***Weng et al., 2010***) | N/A | |
| Strain (*D. melanogaster*) | CadN1-2 Δ14 | T.R. Clandinin (***Prakash et al., 2005***) | N/A | |
| Strain (*D. melanogaster*) | Sema-1aP1 | Bloomington Drosophila Stock Center | RRID:BDSC_11097 | |
| Strain (*D. melanogaster*) | P{10XUAS-IVS-myr::GFP}su(Hw)attP8 | Bloomington Drosophila Stock Center | RRID:BDSC_32196 | |
| Strain (*D. melanogaster*) | P{GMR16H03-lexA}attP40 | Bloomington Drosophila Stock Center | RRID:BDSC_52510 | |
| Strain (*D. melanogaster*) | 11–164 GAL4 | Gift from U. Heberlein (***Nern et al., 2008***) | N/A | |
| Strain (*D. melanogaster*) | UAS-dFezf-3xHA | C.Y. Lee (***Janssens et al., 2014***) | N/A | |
| Strain (*D. melanogaster*) | 27G05-FLP (X) | Janelia Farm Research Campus | N/A | |
| Strain (*D. melanogaster*) | P{10XUAS(FRT.stop) GFP.Myr}su(Hw)attP5 | Bloomington Drosophila Stock Center | RRID:BDSC_55810 | |
| Strain (*D. melanogaster*) | 6–60 GAL4 | Gift from U. Heberlein (***Nern et al., 2008***) | N/A | |
| Strain (*D. melanogaster*) | P{UAS-Dcr-2.D}1 | Bloomington Drosophila Stock Center | RRID:BDSC_24648 | |
| Strain (*D. melanogaster*) | 22E09-LexA | Janelia Farm Research Campus | N/A | |
| Strain (*D. melanogaster*) | P{GMR24C08-lexA}attP40 | Bloomington Drosophila Stock Center | RRID:BDSC_62012 | |

*Continued on next page*

*Continued*

| Reagent type (species) | Designation | Source or reference | Identifiers | Add. info. |
|---|---|---|---|---|
| Strain (D. melanogaster) | P{GMR9B08-GAL4}attP2 | Bloomington Drosophila Stock Center | RRID:BDSC_41369 | |
| Strain (D. melanogaster) | P{TRiP.JF02342}attP2 | Bloomington Drosophila Stock Center | RRID:BDSC_26778 | |
| Strain (D. melanogaster) | P{20XUAS-RSR.PEST}attP2 | Bloomington Drosophila Stock Center | RRID:BDSC_55795 | |
| Strain (D. melanogaster) | 18B02-dFezf-C1-3xFLAG | this paper | N/A | See details in 'Construction of transgenic animals' session in Materials and methods |
| Strain (D. melanogaster) | UAS-H2A-GFP | S.L. Zipursky (*Tan et al., 2015*) | N/A | |
| Strain (D. melanogaster) | P{Rh6-EGFP.P}2 | Bloomington Drosophila Stock Center | RRID:BDSC_7461 | |
| Strain (D. melanogaster) | NetBCPTI-000168 | Kyoto Stock Center | RRID:DGGR_115011 | |
| Strain (D. melanogaster) | 18B02-dFezf | this paper | N/A | See details in 'Construction of transgenic animals' session in Materials and methods |
| Strain (D. melanogaster) | Mi{PT-GFSTF.1}dpr 17[MI08707-GFSTF.1] | Bloomington Drosophila Stock Center | RRID:BDSC_61801 | |
| Strain (D. melanogaster) | Mi{PT-GFSTF.1}nrm [MI01630-GFSTF.1] | Bloomington Drosophila Stock Center | RRID:BDSC_60505 | |
| Strain (D. melanogaster) | Mi{PT-GFSTF.0}beat-IIb[MI03102-GFSTF.0] | Bloomington Drosophila Stock Center | RRID:BDSC_59406 | |
| Strain (D. melanogaster) | Mi{PT-GFSTF.1}CG341 13[MI01139-GFSTF.1] | Bloomington Drosophila Stock Center | RRID:BDSC_60162 | |
| Strain (D. melanogaster) | PBac{GMR39D12-lexA}VK00027 | Bloomington Drosophila Stock Center | RRID_BDSC_52681 | |
| Antibody | anti-GFP (chicken) | Abcam | Cat# ab13970; RRID:AB_300798 | 1:1000 |
| Antibody | anti-HA (mouse) | Abcam | Cat# ab1424; RRID:AB_301017 | 1:1000 |
| Antibody | anti-V5 (mouse) | Bio-Rad/AbD Serotec | Cat# MCA2892GA; RRID:AB_1658039 | 1:200 |
| Antibody | anti-HA (rabbit) | Cell Signaling Technologies | Cat# 3724S; RRID:AB_1549585 | 1:1000 |
| Antibody | anti-DsRed (rabbit) | Clontech Laboratories, Inc. | Cat# 632496; RRID:AB_10013483 | 1:200 |
| Antibody | anti-chaoptin (mouse) | Developmental Studies Hybridoma Bank | Cat# 24B10; RRID:AB_528161 | 1:20 |
| Antibody | anti-elav (rat) | Developmental Studies Hybridoma Bank | Cat# 7E8A10; RRID:AB_528218 | 1:200 |
| Antibody | anti-CadN (rat) | Developmental Studies Hybridoma Bank | Cat# DN-Ex 8; RRID:AB_528121 | 1:20 |
| Antibody | anti-FLAG (mouse) | Sigma-Aldrich | Cat# F1804; RRID:AB_262044 | 1:1000 |
| Antibody | Goat anti-chicken IgG (H + L) Alexa Fluor 488 | Thermo Fisher Scientific | Cat# A-11039; RRID:AB_142924 | 1:500 |
| Antibody | Goat anti-rabbit IgG (H + L) Alexa Fluor 568 | Thermo Fisher Scientific | Cat# A-11011; RRID:AB_143157 | 1:500 |
| Antibody | Goat anti-mouse IgG (H + L) Alexa Fluor 647 | Thermo Fisher Scientific | Cat# A-21236; RRID:AB_141725 | 1:500 |
| Antibody | Goat anti-mouse IgG1 Alexa Fluor 568 | Thermo Fisher Scientific | Cat# A-21124; RRID:AB_2535766 | 1:500 |

*Continued on next page*

*Continued*

| Reagent type (species) | Designation | Source or reference | Identifiers | Add. info. |
|---|---|---|---|---|
| Antibody | Goat anti-guinea pig IgG (H + L) Alexa Fluor 568 | Thermo Fisher Scientific | Cat# A-11075; RRID:AB_2534119 | 1:500 |
| Antibody | Goat anti-Rat IgG (H + L) Alexa Fluor 647 | Thermo Fisher Scientific | Cat# A-21247; RRID:AB_141778 | 1:500 |
| Antibody | anti-dFezf (rabbit) | C. Y. Lee (Janssens et al., 2014) | RRID:AB_2568138 | 1:50 |
| Antibody | anti-svp (mouse) | Developmental Studies Hybridoma Bank | RRID:AB_2618079 | 1:10 |
| Antibody | anti-bsh (guinea pig) | S.L. Zipursky (*Hasegawa et al., 2011*) | RRID:AB_2567934 | 1:200 |
| Antibody | anti-pdm3 (guinea pig) | J.R. Carlson (*Tichy et al., 2008*) | RRID:AB_2569865 | 1:500 |
| Antibody | anti-bab1 (rabbit) | S.B. Carroll (*Williams et al., 2008*) | RRID:AB_2570113 | 1:200 |
| Antibody | anti-NetB (guinea pig) | B. Altenhein | N/A | 1:50 |
| Commercial assay or kit | Illumina Nextera XT DNA Library Preparation Kit | Illumina | Cat# FC-131–1024 | |
| Commercial assay or kit | Illumina Nextera XT Index Kit | Illumina | Cat# FC-131–1001 | |

## Full genotypes of flies in each experiment

| Relevant figures | Abbreviated names | Genotypes |
|---|---|---|
| *Figure 2* | | |
| A-D | | w;;9–9 GAL4, UAS-myr::GFP |
| E, F, K, M | wt | w; tubP-GAL80, FRT40A, 27G05-FLP/FRT40A, LexAop-myr::tdTomato; 9–9 GAL4, UAS-R/ 79C23S-GS-RSRT-STOP -RSRT-smFP_V5-2A-LexA |
| G, H, I, L, N, O, P, Q, | dFezf[1] | w; tubP-GAL80, FRT40A, 27G05-FLP/ dFezf[1], FRT40A, LexAop-myr::tdTomato; 9–9 GAL4, UAS-R/ 79C23S-GS-RSRT-STOP -RSRT-smFP_V5-2A-LexA |
| *Figure 3* | | |
| A, B, F, G, N | wt | w; tubP-GAL80, FRT40A, 27G05-FLP/ FRT40A; 9–9 GAL4, UAS-myr::GFP |
| C, D, H, I, J, O | dFezf[1] | w; tubP-GAL80, FRT40A, 27G05-FLP/ dFezf[1], FRT40A; 9–9 GAL4, UAS-myr::GFP |
| M | dFezf[1] + BAC | w; Tub-GAL80, FRT40, 27G05-FLP::PEST/ dFezf[1], FRT40; 9–9 GAL4, UAS-myr::GFP /18B02-dFezf (BAC) |
| P | CadN, Sema-1a, dFezf[1] | w; tubP-GAL80, FRT40A, 27G05-FLP/ CadN1-2 Δ14, Sema-1a P1, dFezf[1], FRT40A; 9–9 GAL4, UAS-myr::GFP |
| Q | CadN, Sema-1a | w; tubP-GAL80, FRT40A, 27G05-FLP/ CadN1-2 Δ14, Sema-1a P1, FRT40A; 9–9 GAL4, UAS-myr::GFP |
| *Figure 4* | | |

*Continued on next page*

*Continued*

| Relevant figures | Abbreviated names | Genotypes |
|---|---|---|
| A, C | mis-exp | 27G05-FLP; UAS-FRT-STOP-FRT-myr::GFP; 6–60 GAL4/UAS-dFezf-3xHA |
| B | wt | 27G05-FLP; UAS-FRT-STOP-FRT-myr::GFP; 6–60 GAL4 |
| E-E''', G, I | mis-exp | UAS-myr::GFP; tubP-GAL80, FRT40A, 27G05-FLP/16H03-LexA, FRT40A, LexAop-myr::tdTomato; 11–164 GAL4/UAS-dFezf-3xHA |
| F, H | wt | UAS-myr::GFP; tubP-GAL80, FRT40A, 27G05-FLP/16H03-LexA, FRT40A, LexAop-myr::tdTomato; 11–164 GAL4 |
| *Figure 5* | | |
| A, B, C, D | | w; tubP-GAL80, FRT40A, 27G05-FLP/FRT40A (FRT40A, *dFezf1*); 9–9 GAL4, UAS-myr::td Tomato/UAS-H2A-GFP |
| *Figure 6* | | |
| A, B, E | | NetB[CPTI-000168]/w(Y); Bl/CyO-GFP; 18B02 -dFezf-C1-3xFLAG (BAC)/TM2(TM6B) |
| C | | NetB[CPTI-000168]/w(Y); FRT40, LexAop-myr::tdTomato/Bl; 31C06-LexA, UAS-dFezf-HA/TM6B |
| D, D' | | NetB[CPTI-000168]/w(Y); 16H03-LexA, LexAop-myr::tdTomato/Bl(CyO-GFP); TM2/TM6B |
| F | | NetB[CPTI-000168]/w(Y); FRT40, LexAop-myr::tdTomato/Bl; 31C06-LexA, UAS-d Fezf-HA/TM6B |
| G, H | | NetB[CPTI-000168]/w; Bl/Sco; UAS-d Fezf-HA/TM6B |
| *Figure 7* | | |
| A, A' | wt | w; tubP-GAL80, FRT40A, 27G05-FLP/FRT40A, LexAop-myr::tdTomato; 9–9 GAL4, UAS-R/ 79C23S-G5-RSRT-STOP-RSRT-SmGFP 10xV5-2A-LexA |
| B, B' | dFezf[1] | w; tubP-GAL80, FRT40A, 27G05-FLP/ dFezf[1], FRT40A, LexAop-myr::tdTomato; 9–9 GAL4, UAS-R/79C23S-G5-RSRT-STOP -RSRT-SmGFP10xV5-2A-LexA |
| C, E | control | UAS-Dcr2 (w)/yv (w, or Y); 22E09-LexA, LexAop-myr::tdTomato/+; TM6B/UAS-dFezf RNAi (BDSC 26778) |
| D, F | dFezf RNAi | UAS-Dcr2 (w)/yv (w, or Y); 22E09-LexA, LexAop-myr::tdTomato/+; 9B08-GAL4 (TM6B) /UAS-dFezf RNAi (BDSC 26778) |
| G, I | control | NetB[CPTI-000168]/w; Bl/Sco; UAS-dFezf-HA/TM6B |
| H, J | mis-expression | NetB[CPTI-000168]/w; Bl/Sco; UAS-dFezf-HA/11–164 GAL4 |
| K | control | UAS-Dcr2/yv (w, or Y); Rh6-EGFP/22E09-LexA, LexAop-myr::tdTomato; UAS-dFezf RNAi (BDSC 26778) (TM6B)/9B08-GAL4 (TM2) |
| L | dFezf RNAi | UAS-Dcr2/yv (w, or Y); Rh6-EGFP/22E09-LexA, LexAop-myr::tdTomato; UAS-dFezf RNAi (BDSC 26778)/9B08-GAL4 |
| *Figure 8* | | |

*Continued on next page*

*Continued*

| Relevant figures | Abbreviated names | Genotypes |
|---|---|---|
| A-B'', C, D-D', E-E'', F-F' | wt | w/w(Y); FRT40/Tub-GAL80, 27G05-Flp, FRT40; 9–9 GAL4, UAS-R/24C08-LexA, LexAop-myr::td-Tomato |
| G-G', H-H', I-I' | dFezf[1] dFezf2 | w/w(Y); dFezf[1] (dFezf[2]), FRT40/Tub-GAL80, 27G05-FLP, FRT40; 9–9 GAL4, UAS-R/24C08-LexA, LexAop-myr::td-Tomato |
| *Figure 3—figure supplement 1* | | |
| A-C | dFezf[2] | w; Tub-GAL80, FRT40, 27G05-FLP::PEST / dFezf[2], FRT40; 9–9 GAL4, UAS-myr::GFP/ TM2(TM6B) |
| *Figure 3—figure supplement 2* | | |
| A, C, | dFezf[1] + BAC | w; Tub-GAL80, FRT40, 27G05-FLP::PEST/ dFezf[1], FRT40; 9–9 GAL4, UAS-myr::GFP/ 18B02-dFezf (BAC) |
| E | dFezf[2] + BAC | w; Tub-GAL80, FRT40, 27G05-FLP::PEST/ dFezf[2], FRT40; 9–9 GAL4, UAS-myr::GFP/ 18B02-dFezf (BAC) |
| *Figure 4—figure supplement 1* | | |
| A | dFezf mis-exp (24 hr APF) | 27G05-FLP; UAS-FRT-STOP-FRT-myr::GFP; 6–60 GAL4/UAS-dFezf-3xHA |
| B and D | wt | UAS-myr::GFP/27G05-FLP(attp18); tub-GAL80, FRT40A/FRT40A, LexAop-myr::tdTomato; 11–164 GAL4/ 39D12-LexA |
| C and D | mis-exp | UAS-myr::GFP/27G05-FLP(attp18); tub-GAL80, FRT40A/FRT40A, LexAop-myr::tdTomato; 11–164 GAL4/ 39D12-LexA, UAS-dFezf-3xHA |
| *Figure 5—figure supplement 1* | | |
| A, A' | wt | w; tubP-GAL80, FRT40A, 27G05-FLP/FRT40A; 9–9 GAL4, UAS-myr::tdTomato/Mi{PT-GFSTF.1} dpr17[MI08707-GFSTF.1] |
| B, B' | dFezf[1] | w; tubP-GAL80, FRT40A, 27G05-FLP/dFezf1, FRT40A; 9–9 GAL4, UAS-myr::tdTomato/Mi{PT-GFSTF.1} dpr17[MI08707-GFSTF.1] |
| C, C' | wt | w; tubP-GAL80, FRT40A, 27G05-FLP/FRT40A; 9–9 GAL4, UAS-myr::tdTomato/Mi{PT-GFSTF.1} nrm[MI01630-GFSTF.1] |
| D, D' | dFezf[1] | w; tubP-GAL80, FRT40A, 27G05-FLP/dFezf1, FRT40A; 9–9 GAL4, UAS-myr::tdTomato/Mi{PT-GFSTF.1}nrm [MI01630-GFSTF.1] |
| E, E' | wt | w; tubP-GAL80, FRT40A, 27G05-FLP/ FRT40A; 9–9 GAL4, UAS-myr::tdTomato/ Mi{PT-GFSTF.0}beat-IIb[MI03102-GFSTF.0] |
| F, F' | dFezf[1] | w; tubP-GAL80, FRT40A, 27G05-FLP/dFezf1, FRT40A; 9–9 GAL4, UAS-myr::tdTomato/Mi{PT-GFSTF.0} beat-IIb[MI03102-GFSTF.0] |
| G, G' | wt | w; tubP-GAL80, FRT40A, 27G05-FLP/FRT40A; 9–9 GAL4, UAS-myr::tdTomato/Mi{PT-GFSTF.1} CG34113[MI01139-GFSTF.1] |
| H, H' | dFezf[1] | w; tubP-GAL80, FRT40A, 27G05-FLP/dFezf1, FRT40A; 9–9 GAL4, UAS-myr::tdTomato/Mi{PT-GFSTF.1} CG34113[MI01139-GFSTF.1] |
| *Figure 5—figure supplement 3* | | |

*Continued on next page*

*Continued*

| Relevant figures | Abbreviated names | Genotypes |
|---|---|---|
| A, B | wt | w; tubP-GAL80, FRT40A, 27G05-FLP/FRT40A; 9-9-GAL4, UAS-myr::tdTomato/UAS-H2A-GFP |
| A, B | dFezf[1] | w; tubP-GAL80, FRT40A, 27G05-FLP/dFezf1, FRT40A; 9-9-GAL4, UAS-myr::tdTomato/UAS-H2A-GFP |
| C | CadN, Sema-1a | w; tubP-GAL80, FRT40A, 27G05-FLP/CadN1-2 Δ14, Sema-1a P1, FRT40A, LexAop-myr::tdTomato; 9-9-GAL4, UAS-R/79C23S-GS-RSRT-STOP-RSRT-smFP_V5-2A-LexA |
| C | wt | w; tubP-GAL80, FRT40A, 27G05-FLP/FRT40A, LexAop-myr::tdTomato; 9-9-GAL4, UAS-R/79C23S-GS-RSRT-STOP-RSRT-smFP_V5-2A-LexA |
| C | dFezf[1] | w; tubP-GAL80, FRT40A, 27G05-FLP/dFezf1, FRT40A, LexAop-myr::tdTomato; 9-9-GAL4, UAS-R/79C23S-GS-RSRT-STOP-RSRT-smFP_V5-2A-LexA |
| *Figure 6—figure supplement 1* | | |
| A, B | | w; Bl/CyO-GFP; 18B02-dFezf -C1-3xFLAG (BAC)/TM6B |

## Experimental model and subject details

### Description of replicates

In general, the replicates for each experiment are indicated by the number of cells and brains examined (denoted in the figures and figure legends). There are essentially three types of replicates we consider. Within each optic lobe, each column is replicated hundreds of times, different lobes of the same brain are replicates, and the lobes of different brains are also replicates. In all experiments, the phenotypes reported are consistently reproducible for each type of replicate considered (i.e. within and between different animals)

### Fly strains

Flies were raised on standard cornmeal-agar based medium. Male and females flies were used at the following development stages: 3[rd] instar, 12 hr after pupariam formation (h APF), 24 hr APF, 48 hr APF, 72 hr APF and newly eclosed adults (within 5 hr of eclosion).

The following strains were obtained from Bloomington Stock Center (Indiana University):

10xUAS-IVS-myr::GFP [attp2], tubP-GAL80, FRT40A, R27G05-FLPG5.PEST [attP40], 20XUAS-RSR.PEST [attP2], 10XUAS-IVS-myr::GFP [su(Hw)attP8], GMR16H03-lexA [attP40], 10xUAS-FRT-stop-FRT-myr::GFP [su(Hw)attP5], UAS-Dcr-2, GMR24C08-lexA [attP40], GMR9B08-GAL4 [attP2], TRiP.JF02342 [attP2] (dFezf RNAi), 20xUAS-RSR.PEST [attP2], Rh6-EGFP

The following strains were obtained from Janelia Research Campus:

R27G05-FLPG5.PEST [attp18], 22E09-LexA, 39D12-LexA

The following stocks were generated in the Pecot Laboratory for this study:

18B02-dFezf, 18B02-dFezf-C1-3xFLAG

The following strains were gifts from other laboratories:

9–9 GAL4, 6–60 GAL4, 11–164 GAL4 (U. Heberlein), LexAop-myr::tdTomato [su(Hw)attP5] (S.L. Zipursky), UAS-dFezf-3xHA, dFezf[1], dFezf[2] (C.Y. Lee), CadN1-2 [Δ14] (T.R. Clandinin)

## Construction of transgenic animals

### Generation of the BRP presynaptic marker 79C23S-RSRT-STOP-RSRT-smFPV5-2A-LexA

Restriction free cloning (van den Ent and Löwe 2006) was used to insert smFP_V5 (*Viswanathan et al., 2015*) (addgene: pCAG_smFPV5 plasmid# 59758) in between RSR and the 2A peptide (2A) within 'cassette F' (see below), which was made by Chen and colleagues for generation of the original Brp STaR marker (*Chen et al., 2014*). This replaced the original V5 tag and generated:

Cassette G: GS linker-RSR-STOP-RSR-smFP_V5-2A-LexAVP16

[Cassette F: GS linker-RSR-STOP-RSR-V5-2A-LexAVP16]

The modification of the BAC CH321-79C23S (*Chen et al., 2014*) was performed using the recombineering protocol described previously (*Sharan et al., 2009*).

### Generation of 18B02-dFezf and 18B02-dFezf-C1-3xFLAG

A bacterial artificial chromosome (BAC) (CH321-18B02) containing the dFezf locus was ordered from BACPAC Resources, and inserted into the VK33 genomic site, resulting in the 18B02-dFezf strain.

The modification of the BAC was performed using the recombineering protocol described previously (Sharan et al., 2009, Nature Protocols).

The wild type 18B02 BAC was transformed into the SW102 E. coli strain (*Warming et al., 2005*). RpsL-Kan cassette with homology arms flanking the first stop codon of the dFezf locus was amplified using the following primers:

Forward:           ACCATCAGCAGCAGCAAAGACTCTCGGAGACCTTCATAGCCAAGGTGTTTAGCTTCACGCTGCCGCAAGCACTCAG

Reverse:           GAGGTCGACCCCGTGGAGCTGTTCAAGCTTTCCGCATAGTATCGCTGTCAGGGGTGGGCGAAGAACTCCAGCATGA

The PCR product was transformed into the 18B02-harboring SW102 cells, and the RpsL-Kan cassette was inserted before the first stop codon by homologous recombination.

1 kb of DNA fragment centered by the first stop codon of the dFezf locus was PCR amplified from the genomic DNA using the following primers:

Forward: CTCACCTTCCACATGCACAC

Reverse: GCGTGTCTACACGGAACTCA

This fragment was then cloned into pGEM-T vector (Promega, Madison, USA). Using the resulting construct as template, site-directed mutagenesis was performed by PCR using the following primers so that the 3xFLAG sequence was inserted before the first stop codon of the dFezf locus on the plasmid:

Forward:   GAGACCTTCATAGCCAAGGTGTTTGACTACAAAGACCATGACGGTGATTATAAAGATCATGACATCGATTACAAGGATGACGATGACAAGTGACAGCGATACTATGCGGAAAGC

Reverse: CGAGAGTCTTTGCTGCTGCTGATG

Using the resulting construct as the template, PCR was performed using the following primers to amplify the DNA fragment having 3xFLAG inserted before the first stop codon of the dFezf locus and flanked by ~500 bp of homology arms:

Forward: CATCCGCAAAGCCAGCAACA

Reverse: GAGGTGGCCATCGAGGATAT

The PCR product was then transformed into the SW102 cells that harbored modified 18B02 with RpsL-Kan cassette inserted before the first stop codon of the dFezf locus. The RpsL-Kan cassette was then replaced by 3xFLAG by recombination, resulting in a BAC that has 3xFLAG inserted right before the first stop codon of the dFezf locus. The 3xFLAG tagged BAC was inserted into the VK33 genomic site, resulting in the 18B02-dFezf-C1-3xFLAG strain.

## Method details

### Immunohistochemistry

Fly brains were dissected in Schneider's medium and fixed in 4% paraformaldehyde in phosphate buffered lysine for 25 min. After fixation, brains were quickly washed with phosphate buffer saline (PBS) with 0.5% Triton-X-100 (PBT) and incubated in PBT for at least 2 hr at room temperature. Next, brains were incubated in blocking buffer (10% NGS, 0.5% Triton-X-100 in PBS) overnight

at 4°C . Brains were then incubated in primary antibody (diluted in blocking buffer) at 4°C for at least two nights. Following primary antibody incubation, brains were washed with PBT three times, 1 hr per wash. Next, brains were incubated in secondary antibody (diluted in blocking buffer) at 4°C for at least two nights. Following secondary antibody incubation, brains were washed with PBT two times, followed by one wash in PBS, 1 hr per wash. Finally, brains were mounted in SlowFade Gold antifade reagent (Thermo Fisher Scientific, Waltham, MA).

Confocal imaging was accomplished using either a Leica SP8 laser scanning confocal microscope or an Olympus FV1200 Laser Scanning Microscope. Fiji (*Schindelin et al., 2012*; *Schneider et al., 2012*) was used to create z stack images.

The primary antibodies used were as follows: anti-GFP (chicken, 1:1000, ab13970) and anti-HA (mouse, 1:1000, ab1424) were purchased from Abcam. Anti-V5 (mouse, 1:200, MCA2892GA) was purchased from Bio-Rad. Anti-HA (rabbit, 1:1000, 3724S) was purchased from Cell Signaling Technologies. Anti-DsRed (rabbit, 1:200, 632496) was purchased from Clontech Laboratories, Inc. Anti-chaoptin (mouse, 1:20, 24B10), anti-elav (rat, 1:200, 7E8A10), anti-CadN (rat, 1:20, DN-Ex 8) and anti-seven-up (mouse, 1:10, 2D3) were purchased from Developmental Studies Hybridoma Bank. Anti-FLAG (mouse, 1:1000, F1804) was purchased from Sigma-Aldrich. Anti-dFezf (rabbit, 1:100) was a gift from C.Y. Lee. Anti-bsh (guinea pig, 1:200) was a gift from S.L. Zipursky. Anti-pdm3 (guinea pig, 1:500) was a gift from J.R. Carlson. Anti-bab1 (rabbit, 1:200) was a gift from S.B. Carroll. Anti-NetB (guinea pig, 1:50) was a gift from B. Altenhein.

The secondary antibodies used were as follows: Goat anti-chicken IgG (H + L) Alexa Fluor 488 (A-11039), Goat anti-rabbit IgG (H + L) Alexa Fluor 568 (A-11011), Goat anti-mouse IgG (H + L) Alexa Fluor 647 (A-21236), Goat anti-mouse IgG1 Alexa Fluor 568 (A-21124), Goat anti-guinea pig IgG (H + L) Alexa Fluor 568 (A-11075), Goat anti-Rat IgG (H + L) Alexa Fluor 647 (A-21247) were all purchased from Thermo Fischer Scientific and were all used at a dilution of 1:500.

## MARCM and MARCM + STaR experiments

In most MARCM experiments, Flp recombinase was expressed in lamina neuron precursor cells using R27G05-FLPG5.PEST (attP40) to induce mitotic recombination, generating single lamina neuron clones, which were visualized using cell-specific GAL4 drivers. To generate sparse lamina neuron clones in L2 mis-expression experiments (*Figure 4—figure supplement 1B-D*), R27G05-FLPG5.PEST (attp18) was used to induce mitotic recombination.

In experiments utilizing STaR and MARCM Flp recombinase in lamina neuron precursor cells induced mitotic recombination generating a subset of L3 neurons that were homozygous for the control chromosome (FRT40) or *dFezf[1]*. In L3 clones R recombinase was expressed (9–9 GAL4) and induced recombination within the Brp locus in a bacterial artificial chromosome (79C23S) resulting in the incorporation of smFP_V5 at the C-terminus. In addition, this resulted in the co-translation of LexA, which activated the expression of LexAop-myr::tdTomato to visualize the morphology of L3 clones. Together, this allowed visualization of L3 morphology and endogenous pre-synaptic sites with single cell resolution in wild type conditions or when *dFezf* was disrupted, in an otherwise normal fly. smFP_V5 contains ten V5 epitopes built into a superfolding GFP backbone (Viswanathan et al., 2015). We find that it is considerably more sensitive than the original V5 tag in the STaR Brp constructs.

## FlpOut experiments

To generate sparsely labeled L5 neurons, we used an interruption cassette (UAS-FRT-STOP-FRT-myr::GFP) to drive stochastic expression of myr::GFP with an L5 GAL4 driver (6–60 GAL4). Stochastic expression was accomplished through inefficient expression of Flp recombinase in lamina neurons (R27G05-FLP::PEST [attp18]). For reasons that are unknown, this Flp source results in stochastic recombination allowing for visualization of single neurons.

## RNA seq experiments

For each genotype (control, *dFezf[1]*) 120 crosses were set, with 15 females and seven males in each cross. After three days, each cross was flipped every day for a week. From those vials we picked out Tb-negative white pre-pupae that formed within a one-hour window and placed the pupae in separate vials. GFP-positive pupae were scored under the fluorescence microscope and removed from

the vials. The rest of the pupae were kept at 25°C. After 40 hr, we dissected the staged pupae within an hour in cold Complete Schneider's Medium (CSM: 50 mL of CSM consists of 5 mL of heat inactivated fetal bovin serum, 0.1 mL Insulin, 1 mL PenStrep, 5 mL L-Glutamine, 0.4 mL L-Glutathione and 37.85 mL Schneider's medium, filter setrilized). About 50–100 brains were dissected for each independent replicate experiment. After dissection of the brains, the optic lobes were collected and pooled together in 1x Rinaldini's solution (100 mL Rinaldini's solution consists of 800 mg NaCl, 20 mg KCl, 5 mg $NaH_2PO_4$, 100 mg $NaHCO_3$ and 100 mg Glucose in $H_2O$, filter setrilized) for subsequent dissociation. We used a modified dissociation protocol based on one described previously (*Harzer et al., 2013*). The pooled lobes were washed once with 500 uL 1x Rinaldini's solution and then incubated at 37°C for 15 min in dissociation solution that consists of 300 uL papain (100 U/mL) and 4.1 uL Liberase at a final concentration of 0.18 Wu/mL. After the enzyme digestion, the lobes were washed twice with 500 uL of 1x Rinaldini's solution and then twice with 500 uL of CSM. The lobes were then disrupted in 200 uL of CSM by pipetting up and down until the solution became homogeneous. The cell suspention was then filtered through a 30 um mesh 5 mL FACS tube, and diluted with another 500 uL of CSM. The FACS sorting was done on the BD FACS Aria with a nozzle size of 70 um. GFP and dsRed double positive cells were sorted directly into 300 uL of RLT buffer (Qiagen, RNeasy kit) containing 1:100 beta-mercaptoethanol. Total RNA was purified with Qiagen RNeasy kit and eluted in 15 uL of water. After all the samples from five replicates (each genotype) were available, we concentrated the RNA into 5 uL by speed vac. 1.5 uL of the concentrated RNA was used to make cDNA library by Smart-seq2 as previously described (*Picelli et al., 2014*). The quality control of the libraries was done on the Bioanalyzer by Tapestation. The pooled libraries were then loaded onto a single flow cell and sequenced by NextSeq High Output. Between five and fourteen million mapped reads were obtained for each sample.

## Quantification and statistical analyses

### General

The quantification for immunohistochemistry experiments can be found either within the figure or in the figure legend. In all cases, N refers to the number of cells scored.

### Analysis of RNA seq data

The quality of the sequencing files was initially assessed with the FastQC tool. They were then trimmed to remove contaminant sequences (like polyA tails), adapters and low-quality sequences with cutadapt. These trimmed reads were aligned to the drosophila genome (Flybase relase dmel_r6.15_FB2017_02) using STAR aligner. Alignments were checked using a combination of FastQC, Qualimap, MultiQC and custom tools. Counts of reads aligning to known genes were generated by featureCounts. In parallel, Transcripts Per Million (TPM) measurements per isoform were generated using Salmon. Differential expression at the gene level was called with DESeq2. We used the counts per gene estimated from the Salmon by tximport as input to DESeq2 as quantitating at the isoform level has been shown to produce more accurate results at the gene level. As described in the DESeq2 documentation, the differential expression is assessed using a Wald Test and p-values are adjusted for multiple testing using Benjamini-Hochberg FDR correction.

## Acknowledgements

This research has been aided by grants from the Lefler Center for the Study of Neurodegenerative Disorders (MYP, JP) and the Howard Hughes Medical Institute (Gilliam Fellowship for Advanced Study, IJS). We acknowledge CY Lee and H Wang for providing transgenic flies and the dFezf antibody, respectively. We would also like to thank Victor Barrera of the Harvard Chan Bioinformatics Core, Harvard TH Chan School of Public Health, Boston, MA for assistance with the RNA-seq analysis. VB's work was funded by the Harvard NeuroDiscovery Center. Finally, we acknowledge DD Ginty, Y Chen, G Fishell, CC Harwell, CL Cepko and SS Millard for valuable discussions.

## Additional information

### Funding

| Funder | Grant reference number | Author |
|---|---|---|
| Lefler Center for the Study of Neurological Disorders | | Jing Peng<br>Matthew Y Pecot |
| McKnight Foundation | | Matthew Y Pecot |
| Howard Hughes Medical Institute | Gilliam Fellowship for Advanced Study | Ivan J Santiago |

The funders had no role in study design, data collection and interpretation, or the decision to submit the work for publication.

### Author contributions

Jing Peng, Data curation, Formal analysis, Investigation, Visualization, Writing—review and editing; Ivan J Santiago, Conceptualization, Data curation, Formal analysis, Validation, Investigation, Visualization, Writing—review and editing; Curie Ahn, Data curation, Formal analysis, Investigation; Burak Gur, Formal analysis, Validation, Investigation, Visualization, Writing—review and editing; C Kimberly Tsui, Zhixiao Su, Formal analysis; Chundi Xu, Writing—review and editing; Aziz Karakhanyan, Formal analysis, Visualization; Marion Silies, Supervision, Writing—review and editing; Matthew Y Pecot, Conceptualization, Data curation, Formal analysis, Supervision, Funding acquisition, Validation, Investigation, Visualization, Methodology, Writing—original draft, Project administration, Writing—review and editing

### Author ORCIDs

Matthew Y Pecot [iD] http://orcid.org/0000-0001-8241-8002

### Decision letter and Author response

Decision letter https://doi.org/10.7554/eLife.33962.027
Author response https://doi.org/10.7554/eLife.33962.028

## Additional files

### Supplementary files

• Supplementary file 1. Table containing normalized transcript values for all genes across all samples (regardless of significance)
DOI: https://doi.org/10.7554/eLife.33962.022

• Supplementary file 2. QC and DE gene analyses.
DOI: https://doi.org/10.7554/eLife.33962.023

• Supplementary file 3. Table of differentially expressed genes containing: bio categories, gene family information, normalized transcript values across all samples, fold change, and p-values.
DOI: https://doi.org/10.7554/eLife.33962.024

• Transparent reporting form
DOI: https://doi.org/10.7554/eLife.33962.025

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
