## [Decision Letter]

Thank you for submitting your article "*Drosophila* Fezf coordinates laminar-specific connectivity through cell-intrinsic and cell-extrinsic mechanisms" for consideration by *eLife*. Your article has been reviewed by three peer reviewers, and the evaluation has been overseen by a Reviewing Editor and K VijayRaghavan as the Senior Editor. The reviewers have opted to remain anonymous.

As you see, reviewers complimented the technical quality of the study, but were split in their assessment whether the advance is sufficient for *eLife*. After extensive discussions, we are writing to invite you for a revision that shall address reviewers' critiques as thoroughly as possible. In particular, all reviewers agreed that a full description of the RNA-seq data and thorough analysis, hopefully with more insights, are required for the revised manuscript to cross the threshold of advance for an *eLife* paper.

Summary:

Neural circuits both in vertebrates and invertebrates are frequently arranged in a layered organization, essential for correct function of these circuits. However, identifying the molecules involved in the precise building of this laminar arrangement, and in the formation of layer-specific connections has been a long-standing question in the field. The present work by Santiago and colleagues furthers our understanding of neuropil targeting and neural circuit formation in the medulla of the *Drosophila* optic lobe. Building on previous results and the data presented in this manuscript, the authors show that layer targeting of two different neurons, lamina neuron L3 and photoreceptor R8, is controlled by the *Drosophila* ortholog of the transcription factors Fezf, genes that play an important role in cortical patterning in mammals. dFezf instructs L3 layer targeting specificity from early pupal stages, being necessary both for broad domain specificity and for final M3 layer targeting specificity. Using RNA-seq and genetic approaches, the authors found that dFezf is cell autonomously required for Netrin expression in L3 neurons.

Importantly, the authors determined that the expression of this transcription factor in L3 neurons is required for correct R8 layer targeting, likely through the activation of Netrin expression in L3 neurons. The authors propose that transcription factors modules regulating layer targeting by acting into different neurons can be a common mechanism of neural circuit formation.

Since Fezf controls the expression of Netrin in L3 neurons and Netrin provided by L3 has been shown to be important for R8 growth cone stabilization in the M3 layer (Akin and Zipursky, 2016), the authors assume that Fezf is controlling R8 targeting through Netrin expression in L3 neurons. However, they also point out that Fezf in L3 neurons could be controlling the expression of additional molecules involved in R8 targeting/recognition. They do not go into detail about the possible molecules/mechanisms.

Our editorial discussions raised some important general concerns which need to be addressed. An expectation, perhaps high, from this study was an evaluation of layered connectivity that goes beyond a single pair of neurons that target one particular layer. In addition, some insight into the genetic program that FezF regulates to control L3 targeting also seems important. There are many examples of ligand receptor systems that control layered targeting, and in many cases, prior studies have gone well beyond a description of the anatomy to evaluate the importance of layer-specific targeting for the control of the physiological properties of the connected neurons. The current paper can also go beyond what is appears a limited scope if the following essential revisions are squarely addressed.

Essential revisions:

The paper is well written and all the figures clearly represent the findings. While we are positive about the quality of this work its significance could be made much more compelling if the complete results of the RNA-seq were included in the paper. This would be simplest. Alternatively other methods should be used to document changes in the expression of relevant transcripts.

The question remains, which molecules downstream of dFezf are involved in L3 layer targeting. It is interesting that the authors did not find candidate molecules in the RNA-seq experiments performed at 40h APF, given that it is an important developmental stage for L3 layer targeting. The authors should provide the results of the screen and comment on what they find in the Discussion. (Also, this question could be better addressed with similar RNA-seq experiments performed at different stages throughout development, although this would be outside of the scope of this paper). Specifically, also see subsection “DFezf functions in parallel to CadN and Sema-1a to regulate L3 growth cone targeting”: the single-cell type RNA-seq analysis for L3 neurons is elegantly performed. While, this is technically attractive, yet very limited data are presented in the manuscript.

Subsection “DFezf acts instructively to regulate growth cone targeting to the proximal domain of the outer medulla”: When authors describe the instructive role of Fezf, authors firstly expressed Fezf only in L5, which did not exhibit defects in L5 neurons (innervation to the proximal domains). On the other hand, authors expressed Fezf in all the lamina neurons (pan-lamina neuron GAL4 driver, 11-164-GAL4) and independently labeled L2 neurons, which exhibited mis-innervation defects. These experiments do not directly indicate the instructive role of Fezf for M3 layer innervation. It is still possible that the forced Fezf expression in the lamina neurons except L2 neurons indirectly affect L2 neuron innervation. Authors should explain this possibility. In addition, it is better to express Fezf only in L2 neuron to check the instructive role of Fezf for M3 layer innervation.

Also, in subsection “DFezf is necessary and sufficient for Netrin expression”: Authors described that forced expression of Fezf in L2 neurons is sufficient to induce Net-B expression without showing any images. It is better to present some images or graphs to explain this result.

The authors show that L3 terminal mis-targeting is associated with changes in the localization of presynaptic Brp puncta (Figure 2), suggesting that dFezf mutant L3 neurons have defects not only in layer innervation but in synaptic partner matching. However, this analysis leaves several questions unresolved that would aid in interpreting the results. Are there quantifiable differences in the number of Brp puncta that are associated with dFezf mutant L3 neurons compared to wild type clones? Also, is there additional evidence that would support the conclusion that L3 presynaptic sites in dFezf mutants represent functioning synapses formed with ectopic partners?

The model strongly predicts that the FezF gain of function phenotypes should depend on induction of Netrin expression and thus should be suppressed in a Netrin mutant background. Is this the case?

As the core of this paper is the function of Fezf, authors should present how Fezf works as "a transcription factor" in M3 layer innervation. For instance, whether can Fezf directly bind to the enhancer of Net-B gene? Which gene (gene set) expression is regulated by Fezf (probably RNA-seq data)? Alternatively, other evidence is also okay, but we suggest that the authors provide and discuss the mechanism how Fezf is involved in M3 layer targeting in more detail.

---

## [Author Response]

Essential revisions:The paper is well written and all the figures clearly represent the findings. While we are positive about the quality of this work its significance could be made much more compelling if the complete results of the RNA-seq were included in the paper. This would be simplest. Alternatively other methods should be used to document changes in the expression of relevant transcripts.

We have included the entire RNA-seq data set in the revised version of the paper, and a detailed analysis of the data, which includes discussion of potential dFezf targets that regulate growth cone targeting in the main text and Discussion sections. The data are presented in several new figures. In general, many genes change in their levels of expression in the absence of dFezf function (455 differentially expressed genes). Similar fractions of genes are either up or downregulated, indicating that dFezf function acts to repress or activate gene expression, directly or through intermediate factors (~20 TFs are differentially expressed). Members of the dpr gene family are prominent targets (the top three most significant genes are dprs). We propose a model wherein dFezf regulates growth cone targeting by controlling a program of dpr gene expression. However, given the complexity of the dFezf gene program, addressing which genes regulate growth cone targeting and how dFezf regulates their expression will require considerable effort and further analyses. This was one of the reasons behind our hesitation for including the data.

Figure 5 demonstrates the efficacy of L3 purification from MARCM experiments using FACS, shows a heat map of all differentially expressed genes and their expression levels in all samples, and also includes a table of differentially expressed dpr genes (11 in total).

In the Materials and methods section (Quantification and statistical analyses: Analysis of RNA seq data), we have included a link to an HTML document containing all quality control and differential gene expression analyses performed on the RNA-seq data.

Supplementary file 1 is an excel file containing normalized transcript values for all genes across all samples (regardless of significance).

Supplementary file 2 is an excel file containing information regarding differentially expressed genes: gene name, bio categories, gene family information, normalized transcript values across all samples, fold change, and p-values.

Figure 5—figure supplement 1 shows validation of changes in mRNA expression for 4 genes at the protein level.

Figure 5—figure supplement 2 shows the distribution of differentially expressed genes in specific biological categories. This shows that more than 100 DE genes are cell surface and secreted molecules.

Discussion of the RNA-seq data has been included in the section titled: “DFezf regulates a complex program of cell surface gene expression”.

We also discuss the data in detail in the Discussion, primarily within the section titled “DFezf cell-intrinsically instructs axon target specificity”, which has been revised to incorporate the implications of these findings, but also within the section titled **“**DFezf regulates layer innervation through a cell-extrinsic mechanism”.

The question remains, which molecules downstream of dFezf are involved in L3 layer targeting. It is interesting that the authors did not find candidate molecules in the RNA-seq experiments performed at 40h APF, given that it is an important developmental stage for L3 layer targeting. The authors should provide the results of the screen and comment on what they find in the Discussion. (Also, this question could be better addressed with similar RNA-seq experiments performed at different stages throughout development, although this would be outside of the scope of this paper). Specifically, also see subsection “DFezf functions in parallel to CadN and Sema-1a to regulate L3 growth cone targeting”: the single-cell type RNA-seq analysis for L3 neurons is elegantly performed. While, this is technically attractive, yet very limited data are presented in the manuscript.

Please see the response to the first essential revision comment above, which we feel also addresses the issues brought up here.

Subsection “DFezf acts instructively to regulate growth cone targeting to the proximal domain of the outer medulla”: When authors describe the instructive role of Fezf, authors firstly expressed Fezf only in L5, which did not exhibit defects in L5 neurons (innervation to the proximal domains). On the other hand, authors expressed Fezf in all the lamina neurons (pan-lamina neuron GAL4 driver, 11-164-GAL4) and independently labeled L2 neurons, which exhibited mis-innervation defects. These experiments do not directly indicate the instructive role of Fezf for M3 layer innervation. It is still possible that the forced Fezf expression in the lamina neurons except L2 neurons indirectly affect L2 neuron innervation. Authors should explain this possibility. In addition, it is better to express Fezf only in L2 neuron to check the instructive role of Fezf for M3 layer innervation.Also, in subsection “DFezf is necessary and sufficient for Netrin expression”: Authors described that forced expression of Fezf in L2 neurons is sufficient to induce Net-B expression without showing any images. It is better to present some images or graphs to explain this result.

We apologize for not clearly describing how the L2 mis-expression experiment was done in the main text. Like the reviewer, we were also concerned about non-autonomous effects of dFezf mis-expression. Since there is not an L2-specific driver that is active in early pupal development (none has been published, and we have thus far not discovered one despite considerable effort) we decided to use a pan-LN driver, but use MARCM to generate sparse clones expressing dFezf. This is now clearly described in the text (Results and Discussion sections) and also included in the Materials and methods section in more detail. Briefly, in our experiments (Figure 4) only L2 clones that were isolated in the column (no other lamina neuron clones present) were considered. This eliminates the possibility of non-autonomous effects from dFezf expressing clones in the same column. We also performed additional MARCM experiments using an alternative genetic background to generate an even sparser distribution of L2 clones (weaker source of Flp recombinase). This allowed a significant number of L2 neurons that were isolated in the home column, and also with respect to adjacent columns (i.e. no neighboring lamina neuron clones) to be identified. Assessment of these L2 clones produced similar results (these are shown in Figure 4—figure supplement 1).

The dFezf mis-expression experiments to assess NetB expression were shown in 6G-J (now Figure 7). We have indicated this more clearly in the main text (subsection “DFezf is necessary and sufficient for Netrin expression”, last paragraph).

The authors show that L3 terminal mis-targeting is associated with changes in the localization of presynaptic Brp puncta (Figure 2), suggesting that dFezf mutant L3 neurons have defects not only in layer innervation but in synaptic partner matching. However, this analysis leaves several questions unresolved that would aid in interpreting the results. Are there quantifiable differences in the number of Brp puncta that are associated with dFezf mutant L3 neurons compared to wild type clones? Also, is there additional evidence that would support the conclusion that L3 presynaptic sites in dFezf mutants represent functioning synapses formed with ectopic partners?

We completely agree that the finding of L3 neurons still elaborating presynaptic sites within the wrong layers is provocative and could suggest a change in synaptic specificity. And that additional analyses would indeed help clarify whether this is the case. However, we feel that addressing this issue is beyond the scope of this paper for several reasons:

1) The main conclusions of the paper do not depend on whether L3 synaptic specificity is altered beyond what we already show when dFezf is disrupted. We feel that our data provide strong evidence that connectivity with Tm9 neurons is disrupted in dFezf mutant L3 neurons, which supports the importance of dFezf for proper laminar-specific connectivity. Especially considering that the L3-Tm9 synapse is the only L3 synapse shown to be functionally and behaviorally relevant to vision. Whether, in the absence of dFezf function, L3 forms inappropriate connections addresses the degree to which connectivity is altered, which we feel is an important but separate issue.

2) Rigorously determining if L3 forms active synapses with inappropriate targets in the absence of dFezf function requires extensive and difficult analyses, including EM (or other synaptic mapping experiments, difficult to preform rigorously at the level of light microscopy), and experiments involving optogenetic stimulation and calcium imaging to confirm EM data and demonstrate the functionality of synapses. Realistically, such studies would take at least a year and likely more to complete.

3) We also thought about comparing the number of presynaptic sites formed between wild type and dFezf mutant L3 neurons. However, it’s unclear what would be gained by this analysis. Forming more or less synapses in the absence of dFezf would not provide insight into whether synapses are active, or the cellular partners involved. In addition, differences in numbers could result from changes in gene expression caused by the loss of dFezf or result non-specifically due to an altered extracellular environment.

We have included a sentence in the main text clarifying the limitations of our data (subsection “DFezf is cell autonomously required for L3 layer specificity”) so that we do not overstate our findings.

The model strongly predicts that the FezF gain of function phenotypes should depend on induction of Netrin expression and thus should be suppressed in a Netrin mutant background. Is this the case?

We actually disagree with the reviewer comments in this case. The gain of function phenotypes we show in the paper involve mis-expression of dFezf in L2 neurons. Under these conditions, rather than terminating in the distal domain of the outer medulla in early pupal development, L2 growth cones mis-target to the proximal domain of the outer medulla. In adult animals, the dFezf expressing L2 neurons innervate the L3 target layer M3. Thus mis-expressing dFezf causes a change in broad domain specificity, and layer specificity. Our model predicts that these gain of function phenotypes are due to the cell-intrinsic function of dFezf, rather than the cell-extrinsic function involving Netrin expression. Indeed, L3 neurons and all lamina neuron subtypes innervate the correct layers in a *NetAB* null background (unpublished results).

In L3 neurons, we hypothesize that dFezf (1) represses the expression of cell surface molecules that mediate interactions with target neurons in the distal domain of the outer medulla, and (2) activates the expression of cell surface molecules that mediate interactions with targets in the proximal domain of the outer medulla, thereby directing growth cone position and laminar specificity. When mis-expressed in L2 neurons, we speculate that dFezf functions in the same way thereby altering L2 growth cone position and layer specificity.

As the core of this paper is the function of Fezf, authors should present how Fezf works as "a transcription factor" in M3 layer innervation. For instance, whether can Fezf directly bind to the enhancer of Net-B gene? Which gene (gene set) expression is regulated by Fezf (probably RNA-seq data)? Alternatively, other evidence is also okay, but we suggest that the authors provide and discuss the mechanism how Fezf is involved in M3 layer targeting in more detail.

We agree with the reviewer’s comments and have included the complete RNA seq-data, as well as a discussion of candidate dFezf targets relevant for growth cone targeting (see response to the first essential revisions comment above). Also, a more thorough discussion of how dFezf controls growth cone targeting is presented in the Discussion section (DFezf cell intrinsically instructs axon target specificity).

The genomic regions containing cis-regulatory elements that control *Netrin* expression have not been mapped, and given the size of the *Netrin* locus (more than 100kb) these may be complex. Our preliminary bioinformatics analysis using publicly available databases (e.g. TRANSFAC) did not identify dFezf binding motifs within the *Netrin* locus, but additional genomic studies (in progress) are necessary to rigorously assess whether *Netrin* is a direct or indirect dFezf target.